# Nitrogen fixation sustained productivity in the wake of the Palaeoproterozoic Great Oxygenation Event

Genming Luo[1,2], Christopher K. Junium[3], Gareth Izon[2], Shuhei Ono[2], Nicolas J. Beukes[4], Thomas J. Algeo[1,5,6], Ying Cui[7], Shucheng Xie[1] & Roger E. Summons[2]

The marine nitrogen cycle is dominated by redox-controlled biogeochemical processes and, therefore, is likely to have been revolutionised in response to Earth-surface oxygenation. The details, timing, and trajectory of nitrogen cycle evolution, however, remain elusive. Here we couple nitrogen and carbon isotope records from multiple drillcores through the Rooihoogte–Timeball Hill Formations from across the Carletonville area of the Kaapvaal Craton where the Great Oxygenation Event (GOE) and its aftermath are recorded. Our data reveal that aerobic nitrogen cycling, featuring metabolisms involving nitrogen oxyanions, was well established prior to the GOE and that ammonium may have dominated the dissolved nitrogen inventory. Pronounced signals of diazotrophy imply a stepwise evolution, with a temporary intermediate stage where both ammonium and nitrate may have been scarce. We suggest that the emergence of the modern nitrogen cycle, with metabolic processes that approximate their contemporary balance, was retarded by low environmental oxygen availability.

[1] State Key Laboratory of Biogeology and Environmental Geology, and School of Earth Science, China University of Geosciences, 430074 Wuhan, China. [2] Department of Earth, Atmospheric and Planetary Sciences, Massachusetts Institute of Technology, 77 Massachusetts Avenue, Cambridge, MA 02139, USA. [3] Department of Earth Sciences, Syracuse University, Syracuse, NY 13244, USA. [4] Department of Geology, DST-NRF Centre of Excellence for Integrated Mineral and Energy Resource Analysis, University of Johannesburg, PO Box 524, Auckland Park 2006, South Africa. [5] State Key Laboratory of Geological Processes and Mineral Resources, China University of Geosciences, 430074 Wuhan, China. [6] Department of Geology, University of Cincinnati, Cincinnati, OH 45221-0013, USA. [7] Department of Earth Science, Dartmouth College, HB 6105, Hanover, NH 03755, USA. Correspondence and requests for materials should be addressed to G.L. (email: gmluo@cug.edu.cn) or to R.E.S. (email: rsummons@mit.edu)

Nitrogen is an essential component of amino acids, nucleotides and other compounds central to the cellular metabolism. Although nitrogen is one of the most abundant terrestrial elements—comprising 78% of the Earth's present atmosphere as $N_2$—it can only be biologically assimilated in certain oxidised (e.g., $NO_3^-$) or reduced (e.g., $NH_4^+$) forms. Thus, bioavailable nitrogen is a key nutrient influencing primary productivity, thereby linking global biogeochemical cycles and the Earth's climate[1,2].

The contemporary marine nitrogen cycle, in which nitrate is the dominant bioavailable nitrogen species, is overwhelmingly governed by biological processes[3]. Microbial nitrogen fixation by diazotrophs, reducing atmospheric $N_2$ to ammonia, is the primary source of bioavailable nitrogen and likely had its roots in the Mesoarchaean[4]. In the presence of oxygen, ammonium formed via organic matter remineralisation undergoes a stepwise oxidation to nitrite ($NO_2^-$) and ultimately to nitrate ($NO_3^-$; nitrification), mediated by aerobic ammonium/ammonia-oxidising bacteria (AOB), archaea (AOA) or methanotrophs[5,6]. In the largely oxygenated contemporary ocean, nitrification is quantitative[3]. Once oxidised, however, bioavailable nitrogen can then be lost in oxygen-deficient settings by denitrification (dissimilatory nitrate reduction) and/or anaerobic ammonium oxidation (anammox; ammonium oxidation coupled to nitrite reduction). These are the principal sinks for bioavailable nitrogen in the present-day ocean[3,7].

In the absence of oxygen, ammonium remains stable and can accumulate in deep waters, as exemplified by the anoxic water column of Framvaren Fjord, Norway, which supports ammonium concentrations in excess of 1 mM[8]. It follows, therefore, that in the oxygen-free early Archaean[9,10], the nitrogen cycle would have been relatively simple and vastly different from its contemporary counterpart[3]. Ammonium would have been the predominant form of fixed nitrogen and would have been both assimilated directly and recycled within the water column[11]. Given this intrinsic link between environmental oxygen availability and nitrogen cycling, the timing of the transition from an anaerobic to an aerobic nitrogen cycle (i.e., featuring metabolisms involving nitrogen oxyanions) has been used as a line of evidence to constrain the minimum age of the emergence of oxygenic photosynthesis[12,13].

Although the early Earth's marine nitrogen cycle was almost certainly different from today's, the timing and trajectory of the transition towards a modern nitrogen cycle (the relative importance of the metabolisms in the nitrogen cycle are similar to those in the modern ocean) remains poorly understood. For example, a recent study proposed that the aerobic nitrogen cycle emerged and stabilised in the immediate aftermath of the Great Oxygenation Event (GOE)[14]. This model, however, is not universally accepted, with others favouring a more protracted evolution. Specifically, Fennel et al.[15], and others since[3,16], proposed that the early anaerobic nitrogen cycle evolved to today's aerobic equivalent in a more protracted or stepwise manner. Beginning with an ammonium-dominant stage, extremely oxygen-deficient (<11 μM) conditions are thought to have resulted in less efficient nitrification (versus denitrification), stabilising ammonium as the dominant bioavailable nitrogen species. Ushered in by moderate increases in environmental oxygen availability, the second stage is thought to have featured nitrification in more oxygen-replete surface waters and quantitative denitrification in adjacent oxygen-deficient deep waters. During this stage, both ammonium and nitrogen oxyanions (nitrate and nitrite) were destabilised, instigating a hypothetical nutrient deficiency—a 'nitrogen famine'. Finally, in the third stage, progressive ventilation of the deep oceans led to the establishment of a nitrogen cycle where both nitrate and nitrite were stabilised as dominant bioavailable nitrogen species.

The transition between successive stages in the evolution of the nitrogen cycle was undoubtedly complex and remains poorly known. Despite these difficulties, nitrogen cycling can be tracked by determining the nitrogen isotopic composition of sedimentary organic phases, which can be applied to illuminate processes active in both recent and ancient oceans alike[17,18]. Biologically mediated redox cycling exerts the dominant control on the $\delta^{15}N$ of dissolved nitrogen species, which is ultimately communicated to the geological record after biological assimilation[18]. These principles were recently applied to reveal how the GOE impacted nitrogen cycling ~2300 million years ago (Ma) through the upper Rooihoogte and lowermost Timeball Hill Formations[14].

Building on this initial study, we determined the $\delta^{15}N_{kerogen}$, $\delta^{15}N_{bulk}$ and $\delta^{13}C_{org}$ from an extended stratigraphic succession spanning the upper Rooihoogte to upper Timeball Hill formations in three drillcores (EBA-2, EBA-4 and KEA-4, separated by ~5 km; Methods) from the Carltonville area of South Africa (Fig. 1), which record the first irreversible rise in atmospheric $pO_2$ above $10^{-5}$ present atmospheric levels (PAL)[19,20]. Geochronological constraints (Fig. 1)[21,22] suggest that the uppermost part of the upper Timeball Hill Formation is roughly correlative with the prelude of the oxygen overshoot inferred to have occurred during the Lomagundi event (~2220–2060 Ma)[23–26]. Thus, the investigated succession spans the first increase in atmospheric $pO_2$ from near zero to potentially significantly elevated levels[23], providing a unique opportunity to examine the evolution of the marine nitrogen cycle in response to evolving atmospheric chemistry.

Our study reveals that aerobic nitrogen cycling was established prior to the disappearance of sulphur mass-independent fractionation (S-MIF); however, the evolution towards its modern counterpart was far from simple, negotiating an ephemeral interval of bioavailable nitrogen scarcity. Rather than instigating a nitrogen famine, promoted by rising environmental oxygen availability, diazotrophy was sufficient to replenish bioavailable nitrogen losses and sustain primary productivity in the wake of planetary oxygenation.

## Results

**Chemostratigraphic trends.** With few exceptions, the $\delta^{13}C_{org}$ values from all three of the studied cores range from −36 to −32‰ (Supplementary Table 1), consistent with typical lower Proterozoic values (Figs. 2 and 3)[27]. A modest increase, from −33 to −31‰, is observed between the S-MIF interval and the overlying transitional interval, followed by a negative shift to −36‰ directly after the disappearance of S-MIF (Fig. 2). Above this stratigraphic level, throughout the Timeball Hill Formation, the $\delta^{13}C_{org}$ values are relatively stable at −32.3 ± 0.8‰ (1σ; n = 31; Fig. 3). This pattern of stabilised $\delta^{13}C_{org}$ is similar to that reported by Zerkle et al.[14] (Fig. 3), but differs from the record of Coetzee et al.[28], who reported greater variability. This difference may reflect a slight lithofacies dependency, given that we targeted mainly organic-rich mud rocks and Coetzee et al.[28] analysed a more diverse range of lithologies.

The bulk ($\delta^{15}N_{bulk}$) and kerogen ($\delta^{15}N_{kerogen}$) nitrogen isotope profiles show similar intra- and inter-core stratigraphic trends (Figs. 2 and 3). Kerogen $\delta^{15}N$ values are consistently depleted by 2 to 4‰ compared with $\delta^{15}N_{bulk}$ values for the same sample. The magnitude of this offset is variable and less pronounced in the Timeball Hill Formation relative to the Rooihoogte Formation. In the S-MIF and transitional interval, $\delta^{15}N_{kerogen}$ values vary around +3.8 ± 0.85‰ (1σ; n = 19; Fig. 2). Concurrent with the loss of S-MIF, both $\delta^{15}N_{kerogen}$ and $\delta^{15}N_{bulk}$ increase to +7 and to +9‰ in the lowermost Timeball Hill Formation and then

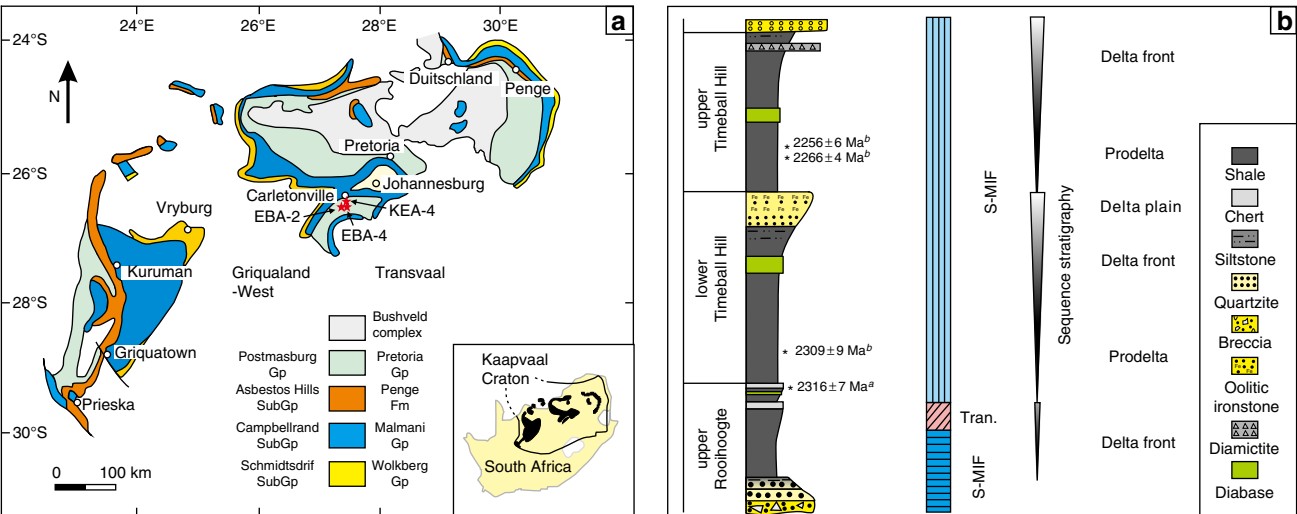

**Fig. 1** Geological map and idealised stratigraphy of the study area. **a** A simplified geological map of the Transvaal Supergroup preserved on the Kaapvaal Craton. **b** Composite stratigraphy of the studied succession in the Carletonville area. The stars in **a** locate the three cores analysed in this study. The geological map is modified from Sumner and Beukes[74]. The generalised lithological column, sequence stratigraphic and facies analysis follow refs. [28, 29]. Note that stratigraphic column is not to scale. Age constraints: [a]Hannah et al.[21], [b]Rasmussen et al.[22]. Quadruple sulphur isotope stratigraphy follows Luo et al.[20]. Gp: group, S-MIF: sulphur mass-independent fractionation, S-MDF: sulphur mass-dependent fractionation, Tran.: transitional interval

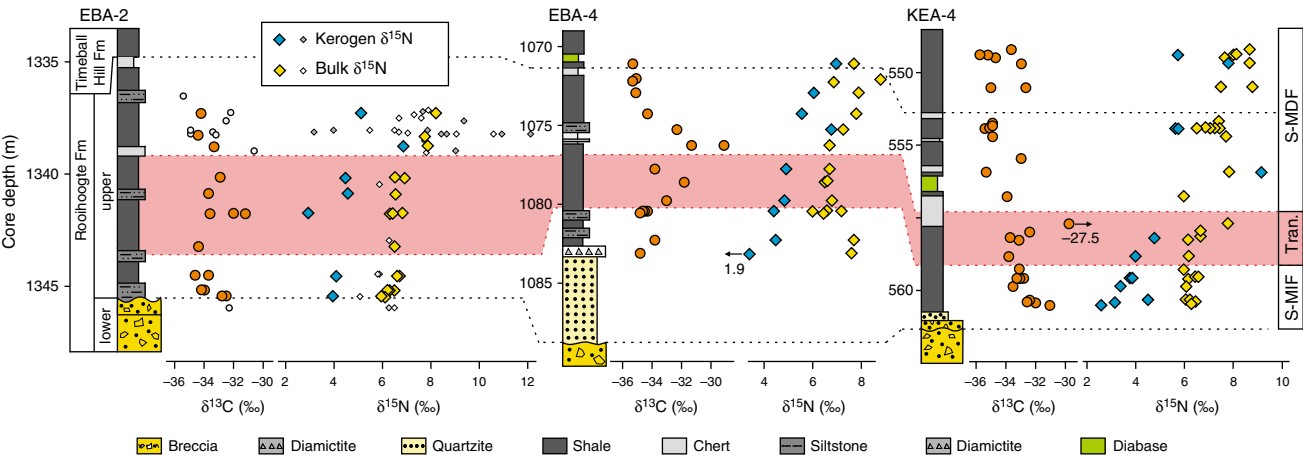

**Fig. 2** Nitrogen and carbon isotope profiles spanning the upper Rooihoogte to lower Timeball Hill formations. Nitrogen and carbon isotopic data are reported relative to atmospheric $N_2$ and VPDB, respectively. The pink band represents the transitional interval (Tran.) from an anoxic atmosphere associated with sulphur mass-independent fractionation (S-MIF) to an oxygenated atmosphere associated with sulphur mass-dependent fractionation (S-MDF)[20]. The smaller symbols represent data from Zerkle et al.[14]

recover to +5 and to +7‰ throughout the remainder of the lower Timeball Hill Formation, respectively (Figs. 2 and 3). Except for one sample in core EBA-4, the transition from the lower to upper Timeball Hill Formation is marked by pronounced ~4‰ negative shifts in both $\delta^{15}N_{bulk}$ and $\delta^{15}N_{kerogen}$, with $\delta^{15}N_{kerogen}$ declining to +1.1‰ ± 1.35‰ (1σ; n = 17). In the uppermost part of the Timeball Hill Formation, a small increase in $\delta^{15}N_{kerogen}$ is observed in cores EBA-2 and EBA-4 but not in core KEA-4, which may reflect sampling biases given the different resolutions of the records.

**The fidelity of the isotope records.** The analysed samples lack obvious evidence for hydrothermal alteration and are surprisingly well preserved given their age[29]. The absence of significant relationship between $\delta^{13}C_{org}$ and total organic carbon (TOC) (Fig. 4a) suggests that thermal alteration is unlikely to have significantly altered the $\delta^{13}C_{org}$ of the studied samples, and the samples lack a significant detrital organic carbon component[30].

Both early and late diagenetic processes have the potential to alter original $\delta^{15}N_{kerogen}$ values[17]. It has been shown that the $\delta^{15}N$ of surface sediments approximates that of sinking organic particles in areas with high sediment accumulation rates, high organic matter content and reducing bottom-water conditions[31]. The expected high sedimentation rates in a deltaic sedimentary environment, and the relatively high TOC content (1.5 ± 1.9 wt.%; 1σ) of the studied samples, suggest that early diagenesis probably had a limited effect on $\delta^{15}N_{kerogen}$. The high $C/N_{kerogen}$ ratios (245 ± 120, n = 75) are similar to those from other Archaean and lower Palaeoproterozoic successions and are consistent with nitrogen loss during mild post-depositional thermal alteration[12,13,32]. Given that the Carletonville strata experienced only lower greenschist metamorphism[28], thermal alteration is not expected to have increased $\delta^{15}N$ by more than~1 to 2‰[33,34]. This inference is braced by the lack of significant correlations between $\delta^{15}N_{kerogen}$ and [$N_{kerogen}$] (Fig. 4b), and $\delta^{15}N_{kerogen}$ and $C/N_{kerogen}$ (Fig. 4c), which is consistent with evidence suggesting

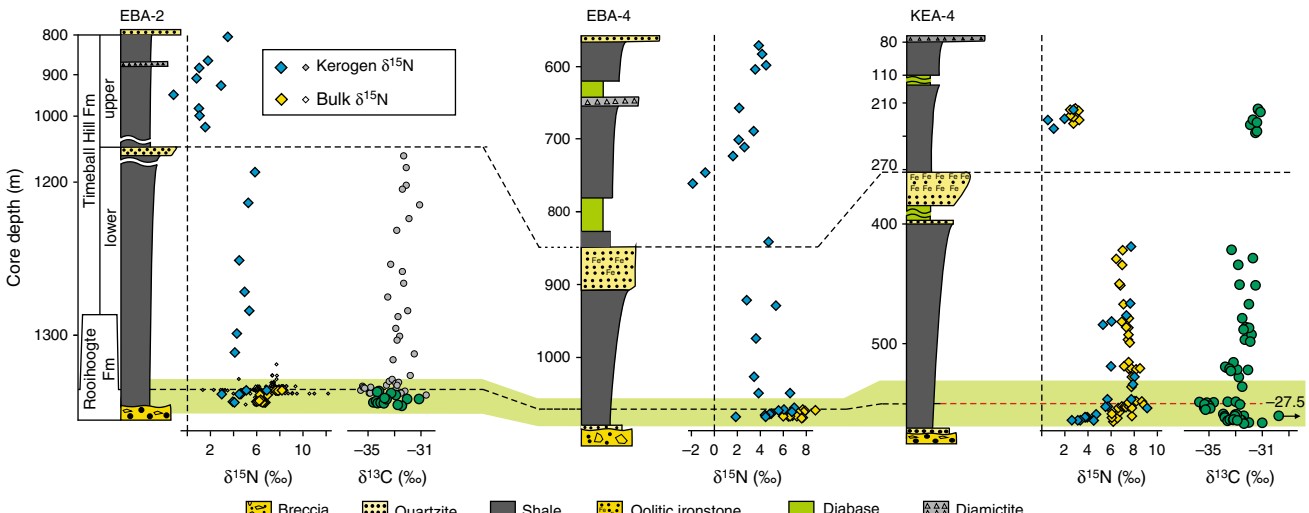

**Fig. 3** Nitrogen and carbon isotope profiles spanning the upper Rooihoogte to upper Timeball Hill formations. Nitrogen and carbon isotopic data are reported relative to atmospheric $N_2$ and VPDB, respectively. Note the differential vertical scale between the lower and upper Timeball Hill Formation in cores KEA-4 and EBA-2. Only the lower part of the upper Timeball Hill Formation in core KEA-4 was analysed in this study. The smaller symbols are data from Zerkle et al.[14] The light green band represents the GOE interval identified by Luo et al.[20]

**Fig. 4** Cross-plots of the elemental and isotopic data. **a** TOC relative to $\delta^{13}C_{org}$, **b** $\delta^{15}N_{kerogen}$ versus nitrogen content in kerogen ($[N]_{kerogen}$ %), **c** $\delta^{15}N_{kerogen}$ relative to C/N atomic ratio of kerogen $(C/N)_{kerogen}$ and **d** $\delta^{13}C_{org}$ versus $\delta^{15}N_{kerogen}$. The data are distinguished by core, with blue diamonds, red circles and yellow squares denoting cores KEA-4, EBA-2 and EBA-4, respectively

that $\delta^{15}N$ values remain largely impervious to heating despite large increases in $C/N_{org}$ ratios[35]. Additionally, high C/N ratios typify black shales throughout the Phanerozoic and are a known by-product of high organic content and diffusion of ammonium out of the sediment during early diagenesis[36].

Bulk sedimentary nitrogen consists not only of organic nitrogen but also of ammonium adsorbed to, and trapped within, the interlayers of clay minerals after substitution for potassium. Generally, ammonium originates from in situ deamination of organic nitrogen, a process that does not result in significant $^{15}N$ fractionation[37]. Clay-hosted ammonium, however, may also have an allochthonous source, originating from either detrital clays or metamorphic/diagenetic fluids. Inclusion of ammoniacal-nitrogen from any of these sources may consequently mute or obscure the primary marine $\delta^{15}N$ signal. In our sample set, interaction with diagenetic fluids and uptake of ammonium onto mineral surfaces/interlayer sites may have lowered the $C_{org}/N_{bulk}$ of some samples; however, this only becomes a major concern when $\delta^{15}N_{bulk}$ deviates significantly from $\delta^{15}N_{kerogen}$ values[38], which is clearly not the case here. Importantly, previous work from core EBA-2 has shown there is no significant correlation between potassium abundance and $\delta^{15}N_{bulk}$, suggesting that any hypothetical nitrogen exchange did not significantly affect the original $\delta^{15}N_{bulk}$ values[14].

Covariation between $\delta^{15}N$ and C/H ratios has recently been used to argue for a thermally driven intra-sample equilibrium effect, which lowers $\delta^{15}N_{Kerogen}$ by ~2‰ in rocks that have experienced lower-greenschist facies metamorphic histories[39]. Consistent with this inference, where we have available coupled $\delta^{15}N_{Kerogen}$ and $\delta^{15}N_{Bulk}$ data, $\Delta^{15}N$ ($=\delta^{15}N_{Kerogen} - \delta^{15}N_{Bulk}$) values vary around $-1.3 \pm 1.3$‰. This close agreement between $\delta^{15}N_{bulk}$ and $\delta^{15}N_{kerogen}$ (Figs. 2 and 3) demonstrates that neither nitrogen pool has experienced significant nitrogen addition, and instead both have experienced typical post-depositional histories, with their respective $\delta^{15}N$ values closely approximating that of the precursor biomass.

## Discussion

Prior to the advent of oxygenic photosynthesis, the Earth's nitrogen cycle operated in an almost entirely anaerobic mode, dominated by diazotrophy and ammonification[17,18]. Diazotrophy is typically accompanied by a small nitrogen isotopic fractionation, whose magnitude is prescribed by the specific nitrogenase responsible for nitrogen fixation (−7 to 0‰)[4,40]. While microbial culture experiments indicate that trace metal availability can alter the magnitude of the nitrogen isotopic fractionation associated with diazotrophy, they demonstrate that nitrogen fixation is unlikely to produce biomass with a $\delta^{15}N$ much higher than 0‰[41,42]. Likewise, remineralisation of diazotrophic biomass carries no appreciable isotope effect ($\sim -1$‰)[18], and simply serves to shuttle $^{15}N$-depleted ammonium to the deep ocean. By contrast, the large fractionation associated with ammonium assimilation (>15‰)[43] produces strongly $^{15}N$-depleted biomass, with $\delta^{15}N$ values lower than the precursor ammonium, as is sometimes suggested for the Archaean record[11,14]. Although ammonium dissociation to ammonia and degassing could conceivably produce positive $\delta^{15}N$ values under an anoxic atmosphere[44], it requires a high pH and is therefore unlikely to have operated outside of atypical and isolated evaporative lacustrine settings[45]. Consequently, the emergence of positive $\delta^{15}N$ values (>3‰) in well-preserved and unaltered sedimentary rocks from open-marine settings is generally interpreted as the isotopic fingerprint of an aerobic nitrogen cycle, constraining when nitrate first formed part of the dissolved nitrogen inventory[11–14].

It is striking that while S-MIF persists into the early Palaeo-proterozoic[9,20,46], Archaean sequences commonly feature positive $\delta^{15}N$ values[11,18]. Although ammonium oxidation to $N_2$ by ferric oxyhydroxides may produce positive $\delta^{15}N$ values, the magnitude of this fractionation remains uncertain[47]. Moreover, it is unlikely to have been of significance in the examined cores given their low ferric oxyhydroxide contents[14]. Therefore, despite the anoxic backdrop, the preferred explanation of these positive $\delta^{15}N$ values commonly invokes the intermittent operation of an aerobic nitrogen cycle[12–14,32]. The sustained positive $\delta^{15}N_{bulk}$ and $\delta^{15}N_{kerogen}$ values preserved in the lower part of all three examined cores (Figs. 2 and 3), in association with pronounced S-MIF[20], imply that nitrification was active and was at least regionally pervasive prior to the accumulation of atmospheric $O_2$ above $10^{-5}$ PAL[10], consistent with the earlier inferences drawn by Zerkle et al.[14] Given the diminishingly low atmospheric oxygen contents necessitated by the persistence of S-MIF (<$10^{-5}$ PAL)[10], and the correspondingly short half-life of oxygen in the highly reducing pre-GOE atmosphere (~9 h)[48,49], aerobic nitrogen cycling, and thus dissolved nitrate availability, must have been spatially variable and confined to sites of nutrient availability and active oxygenic photosynthesis—so-called 'oxygen oases'[50]. Thus, the advent of oxygenic photosynthesis must have significantly predated the GOE, and the positive $\delta^{15}N$ values in the Archaean record might constitute some of the earliest evidence for intermittent biogenic oxygen production[12,13].

In detail, multiple metabolic processes within the aerobic nitrogen cycle produce $^{15}N$-enriched biomass. Nitrification, for example, produces $^{15}N$-depleted nitrate while simultaneously driving the residual ammonium pool to more positive $\delta^{15}N$ values[51–53]. Similarly, denitrification and anammox preferentially shuttle $^{14}N$ to the atmosphere as $N_2$, concomitantly increasing the $\delta^{15}N$ of the residual dissolved nitrogen oxyanion pool[7,54]. In present-day marine settings, the fractionation accompanying nitrification is typically not expressed owing to rapid and quantitative ammonium oxidation to nitrate[7], leaving nitrogen losses, via either denitrification and/or anammox, as the dominant control on the $\delta^{15}N$ of dissolved inorganic nitrogen[7]. By analogy, the positive $\delta^{15}N$ values observed in the prelude to the GOE, reflected in the basal part of our record, have been ascribed to complete nitrification coupled with incomplete denitrification[11–14]. While we acknowledge that most of the processes central to the modern nitrogen cycle (eg, diazotrophy, nitrification and denitrification) had almost certainly evolved by the late Archaean[4,14], their relative importance remains unresolved. This, in turn, leaves an open question: How did the nitrogen cycle evolve in the aftermath of the GOE?

Even directly after the termination of S-MIF, the chemical composition of the early-Palaeoproterozoic ocean–atmosphere system would have remained vastly different from its present-day counterpart, with low atmospheric $pO_2$ and a ferruginous and ammonium-rich deep ocean[55]. At low oxygen concentrations denitrification outpaces nitrification resulting in quantitative loss of nitrite and nitrate. Modelling approaches suggest denitrification outpaced nitrification when dissolved oxygen concentrations were <~11 μm[15], yet the threshold value where ammonium concentrations exceed nitrate concentrations can be substantially higher in contemporary anoxic settings (~35 μm)[8]. Admittedly, evaluating the absolute oxygen contents of the Palaeoproterozoic surface ocean remains difficult; however, dissolved oxygen concentrations were certainly low before and in the immediate aftermath of the GOE, encroaching on the threshold where nitrate/nitrite losses exceeded their replenishment[56–58]. Given these boundary conditions, with a potentially large ammonium reservoir at depth, localised rapid and quantitative denitrification

of a relatively small standing nitrate reservoir may have minimised denitrification's characteristic isotope effect.

Alternatively, the isotope effect associated with nitrification may have been more important than it is today, and offers another explanation for the positive $\delta^{15}N$ values observed in the Carletonville records. In regions where the chemocline was relatively shallow, ammonium may have been the preferred bioavailable nitrogen species, especially given that ammonium suppresses the expression of nitrate reductase and the genes that regulate nitrate assimilation[59]. This explanation, although not universally accepted[45], has been proposed previously to explain the extreme positive $\delta^{15}N$ values recorded in the 2.7 billion-year-old Tumbiana Formation[32]. Given this oxygen-poor backdrop, the nitrogen cycle would have been subtly different and could have been both spatially and diurnally variable. Despite the prevalence of ammonium, in regions of active photosynthetic activity and $O_2$ production, nitrate may have been stable and utilised by phytoplankton as their preferred nitrogen-substrate. Taken together, we argue that, at least initially, the positive $\delta^{15}N$ values seen at and around the termination of S-MIF could have originated from a combination of incomplete nitrification, as well as incomplete denitrification as others have proposed[11–14]. This represents a subtle but distinct difference in the operation of the Palaeoproterozoic nitrogen cycle relative to its present-day counterpart (Fig. 5a). We concede, however, that unequivocally discriminating between incomplete nitrification and denitrification as a source of the positive $\delta^{15}N$ values is impossible. Speculatively, we suggest that the former surrendered to the latter, which attained dominance as the Earth's surface became more oxidised as shown by the stratigraphic $\delta^{15}N$ evolution of the Carletonville records (Figs. 2 and 3).

Throughout the Rooihoogte Formation and up into the lowermost Timeball Hill Formation, $\delta^{15}N_{kerogen}$ values increase from around +2 to +8‰, while $\delta^{15}N_{bulk}$ values increase from around +6 to +9‰ (Fig. 2). These chemostratigraphic trends are accompanied by a decrease in $\delta^{13}C_{org}$ and $\delta^{34}S_{pyrite}$, consistent with increased surface-water oxygen production/availability in turn promoting aerobic nitrogen cycling, oxidation of $^{13}C$-depleted carbon phases (eg, methane or dissolved organic carbon), as well as an increase in the marine sulphate inventory[20]. These coeval isotopic changes argue for a strong yet evolving connection between the biosphere and the chemical composition of the atmosphere and ocean, whereby the chemical evolution of Earth's surficial environments influenced the emergence and distribution of ecological niches, not to mention the metabolisms that filled them. Previously, a negative correlation between $\delta^{13}C_{org}$ and $\delta^{15}N_{kerogen}$ was used to argue that aerobic methanotrophy mediated nitrification in the 2.7 billion-year-old Tumbiana Formation[32]. Aerobic methanotrophy, however, is unlikely to have been important in the Carletonville area over the examined time interval because the $\delta^{13}C_{org}$ values are incompatibly high (>−37‰), and there is no significant correlation between $\delta^{13}C_{org}$ and $\delta^{15}N_{kerogen}$ (Fig. 4d). Therefore, we suggest that AOB and/or AOA were the most likely nitrifiers in this part of the Palaeoproterozoic ocean.

Punctuating the increase in $\delta^{15}N$, in concert with the loss of S-MIF, $\delta^{15}N_{kerogen}$ values are initially extremely variable in core EBA-2 (+1 to +12‰; Fig. 2)[14]; a trend that is mirrored in the other cores, although admittedly at a lower resolution and magnitude (Fig. 2). Given the transiency of the observed $\delta^{15}N$ volatility, the associated nitrogen cycle instability was likely a product of regional water column dynamics and variable redox conditions. Indeed, Fe-speciation data from the basal 20–40 m of core EBA-2 implicate predominantly anoxic (euxinic/ferruginous) depositional conditions, with occasional relaxations toward oxic deposition[14]. Consequently, the decimetre-scale 6–8‰ negative

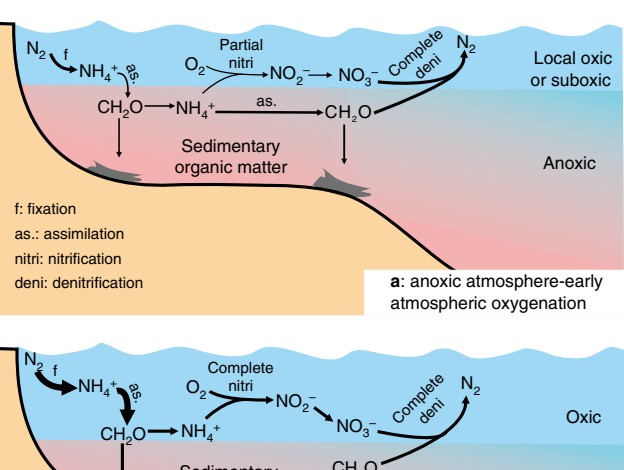

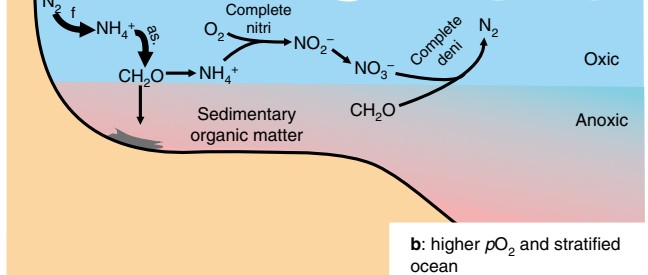

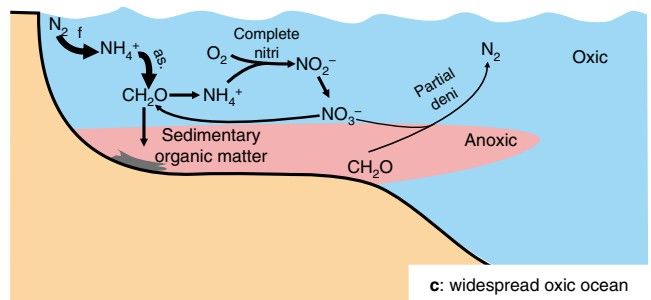

**Fig. 5** Schematic model of the evolving marine nitrogen cycle in response to increasing atmospheric $pO_2$. **a** shows the pre-GOE to early GOE (dissolved $O_2$ content < 11 μM; first stage of Fennel et al.[15]), when the rate of nitrate loss via denitrification exceeded its replenishment via nitrification. Here, ammonium was relatively stable and potentially served as the main source of biologically available nitrogen. **b** depicts the nitrogen cycle after a further increase in environmental oxygen availability (dissolved $O_2$ content > 11 μM; potentially second stage of Fennel et al.[15]), with quantitative nitrification and denitrification destabilising nitrate and ammonium. Microbial nitrogen fixation served as the main source of biologically available nitrogen. **c** depicts the contemporary nitrogen cycle dominated by nitrate in a predominantly oxygenated ocean. We hypothesise that **a** encompasses the upper Rooihoogte Formation to the lower Timeball Hill Formation, **b** encompasses the lower part of the upper Timeball Hill Formation and **c** represents the uppermost Timeball Hill Formation and following Lomagundi event. Geochemical evidence suggests that any rise in atmospheric oxygen was transient and confined to the Lomagundi interval, decreasing in its immediate aftermath[55]. Widespread deep-ocean oxygenation was a much later phenomenon beginning in the Neoproterozoic; therefore, **c** represents a transient state that was not seen for perhaps 1.5 billion years

$\delta^{15}N_{kerogen}$ excursions can be explained by the occasional injection of ammonium-rich anoxic deep-water into the photic zone. Alternatively, denitrification and concomitant nitrogen limitation could have also periodically stimulated diazotrophy, whose isotope effect would have been more pronounced if nitrogen-deficient and $Fe^{2+}$-replete waters were emplaced from depth[41]. Following the initial variability, $\delta^{15}N_{kerogen}$ values stabilise around 5.6 ± 1.5‰ in the lower Timeball Hill Formation (Fig. 3). In this interval, although Fe-speciation data imply localised oxic conditions[14], geochemical proxies (eg, iodine speciation and

selenium isotope ratios)[57,58] and modelling approaches indicate that the global ocean remained stratified with anoxic water beneath oxygen-deficient surface waters (≤10 μM $O_2$)[57]. These data, therefore, require the operation of nitrification, but we are unable to unequivocally decipher whether the positive $\delta^{15}N$ values were a product of an ammonium precursor.

Further up-section, the decrease and stabilisation of $\delta^{15}N$ values to around 0‰ in the upper Timeball Hill Formation (Fig. 3) requires yet another nitrogen cycle adjustment within the Transvaal Bain. Here, the low $\delta^{15}N$ values are consistent with a significant diazotrophic input to the bioavailable nitrogen pool (Fig. 5b). Considering that nitrogen fixation is energetically expensive, these isotopic data imply that the regional bioavailable nitrogen pool had likely been exhausted. In the context of an evolving nitrogen cycle, the transition to diazotrophy in cores separated by upwards of 5 km may provide the first evidence chronicling the inception of widespread nitrogen limitation—a nitrogen famine—driven, perhaps, by redox-controlled bioavailable nitrogen loss, as hypothesised by Fennel et al.[15]. A corollary of the postulated nitrogen famine[15] is that rising oxygen availability prompted quantitative nitrification in the surface ocean as the ocean's interior remained predominantly anoxic[57]. Geochemical data are consistent with these prerequisites, and suggest that dissolved oxygen concentrations were higher during deposition of the upper Timeball Hill Formation relative to its older lithostratigraphic counterpart[26,60,61].

The global expression of nitrogen limitation, and the feasibility of a widespread nitrogen famine, is dependent on the global balance between nitrogen losses via canonical denitrification, its retention as ammonia following dissimilatory nitrate reduction[62], and its replenishment via diazotrophy. The secular evolution of P/Fe ratios suggests that dissolved phosphate remained replete throughout the early Palaeoproterozoic and that non-ferrous trace metals may have been biolimiting[63]. Conceptually, enhanced metal sequestration beneath a euxinic water column, driven by enhanced sulphate availability, was envisaged to have dampened trace metal inventories in the wake of the GOE[64]. Given that molybdenum is an essential enzymatic-cofactor in nitrogen fixation[4], it has been proposed that molybdenum and nitrogen co-limitation could have curtailed biological activity throughout the mid-Proterozoic[64]. Cyanobacterial metal limitation experiments, however, demonstrate that contemporary nitrogen fixation rates can be maintained at vanishingly low molybdenum concentrations (~5 nM)[65,66], which approximate those predicted for Proterozoic oceans[67]. Given that molybdenum concentrations encroached on levels where co-limitation could have occurred, the Mo-N co-limitation hypothesis cannot be unequivocally dismissed. Nevertheless, several lines of evidence suggest that diazotrophy could keep pace with bioavailable nitrogen losses, sustaining primary productivity in this part of the Palaeoproterozoic ocean.

The continued and prodigious release of $O_2$ necessitated by the Lomagundi carbon isotope excursion[23], requires quantitative recycling of organic nitrogen and/or pervasive diazotrophy. Consequently, extrapolating our observations outside of the Carletonville area, we hypothesise that enhanced phosphorous availability—derived from enhanced weathering[68] coupled with intensified redox-promoted recycling[69,70]—associated with evolving ocean–atmospheric chemistry[55,61], was capable of sustaining widespread, and potentially more efficient diazotrophy[16], replenishing the lost nitrate and preventing any long-lived nitrogen famine. The poorly defined positive shift in the uppermost part of the upper Timeball Hill Formation (Fig. 3), in the prelude to the Lomagundi oxygen overshoot (Fig. 1)[22], is also consistent with short-lived nitrate scarcity. Here $\delta^{15}N$ values approximate those of contemporary dissolved nitrate (~4‰),

which tentatively suggest increasing denitrification, consistent with contraction of anoxia and partial oxygenation of the deep ocean[24,25,57,58]. Consequently, we argue that any hypothetical nitrogen famine would have been a transient Earth-system state, and would have most likely been terminated by the onset of the Lomagundi Event (~2220 Ma). Therefore, assuming the decrease in $\delta^{15}N$ values to around 0‰ represents a global nitrogen cycle response, constrains any potential nitrogen famine to ~2250–2220 Ma.

Further increases in atmospheric $pO_2$, during the Lomagundi interval and beyond, would have ultimately stabilised dissolved nitrogen oxyanions and ushered in a nitrogen cycle more reminiscent of the present day (Fig. 5c). This newly plentiful nitrate has been suggested to have catalysed the diversification of nitrate-assimilating cyanobacteria and potentially the emergence of eukaryotes[14]; yet, was apparently insufficient to expedite the proliferation of eukaryotes[71]. High-resolution $\delta^{15}N$ studies, particularly in the prelude and aftermath of the Lomagundi event, are required to test our inferences and more completely explore whether the evolution of the marine nitrogen cycle was uni-directional, or whether it was spatio-temporally dynamic, as might be predicted from consideration of other oxygen sensitive proxy records[55,58]. Coupling more complete $\delta^{15}N$ records with biomarker and molecular clock approaches will further refine our understanding of the ancient nitrogen cycle, providing insight into the emergence and proliferation of specific metabolisms responsible for nitrogen cycling—unveiling the timing, tempo and trajectory of when the nitrogen cycle became ecologically modern.

## Methods

**Samples and geological background**. The samples from the upper Rooihoogte to uppermost Timeball Hill formations were collected from cores KEA-4, EBA-4 and EBA-2, which were drilled in the Carltonville area of South Africa (Fig. 1). Details about the sedimentary environments of this succession can be found in Coetzee[29]. Briefly, the upper Rooihoogte Formation comprises mudstones, siltstones and silicified stromatolites, which were deposited in a delta-front environment[29]. The Timeball Hill Formation contains two upward-coarsening parasequences, each representing a prograding deltaic complex comprised of prodeltaic mudstones overlain by delta-front mudstones. The lower parasequence is bound by a quartzite and oolitic ironstone at its top, while a glacial diamictite delimits the top of the upper parasequence[28,29]. The geochronological framework of the studied succession is well established (Fig. 1). The Rooihoogte–Timeball Hill formation boundary has been dated to 2316 ± 7 Ma using the Re-Os system[21], which is consistent with a 2310 ± 9 Ma zircon U-Pb age for the base of the Timeball Hill Formation (Fig. 1)[22]. Additionally, the lower part of the upper Timeball Hill Formation has been dated to ~2250 Ma (Fig. 1)[22], which is close to the inception of the Lomagundi oxygen overshoot (~2220–2060 Ma)[23–26].

**Kerogen isolation**. Core samples were cut into small pieces, removing all exterior surfaces. Fresh chips were ultrasonically cleaned with successive deionised water, methanol, and dichloromethane rinses prior to homogenisation (<100 mesh) using a precleaned stainless steel puck mill[20]. Aliquots of samples used previously for sulphur isotope analysis[20] were quantitatively decalcified using 6 M HCl before being rinsed to neutrality using ultra-pure water and dried at 60 °C. Kerogens were isolated at Massachusetts Institute of Technology using a method modified from Zerkle et al.[14]. Briefly, the carbonate fraction was removed from ~4 g aliquots of rock powder via an overnight 6 M HCl dissolution. The residues were then rinsed to neutrality using successive ultra-pure water, decanting the supernatant following centrifugation. The silicate fraction was then volatilised via duplicate overnight treatments with 20 ml of a mixed HF and HCl (3:2) solution. Each acid treatment was followed by multiple ($n = 4$) 1 M HCl rinses and then dried for mass spectrometric analysis. The low carbonate abundances and the high thermal maturity negated the need for replicate 6 M HCl and dichloromethane treatments. Heavy minerals such as pyrite were not removed as they do not affect nitrogen isotopic compositions.

**Nitrogen and carbon isotope analysis**. Bulk nitrogen ($\delta^{15}N_{bulk}$) and carbon ($\delta^{13}C_{org}$) isotope values were determined by flash combustion using an elemental-analyser (EA) coupled to a continuous-flow isotope ratio mass spectrometer (IRMS) at Massachusetts Institute of Technology. The low nitrogen content prevented simultaneous $\delta^{15}N_{bulk}$ and $\delta^{13}C_{org}$ determinations. For $\delta^{15}N_{bulk}$ analysis, the decalcified residues (~40 to 80 mg) were combusted at 1040 °C, and the

resulting nitrogen oxides were reduced to $N_2$ by Cu at 650 °C. The analyte gas stream was then stripped of $CO_2$ and water via contact with Carbosorb® and anhydrous magnesium perchlorate, respectively. For $\delta^{13}C_{org}$ analysis, smaller aliquots of decarbonated residues (~1–8 mg) were combusted and dried as detailed for $\delta^{15}N_{bulk}$ determination. Most samples were analysed in triplicate, yielding standard deviations ($1\sigma$) for $\delta^{15}N_{bulk}$ and $\delta^{13}C_{org}$ better than 0.3‰ ($1\sigma$) and 0.2‰ ($1\sigma$), respectively. All isotope data are reported in standard delta-notation, reflecting per mille variations from their respective international reference standards:

$$\delta^a X(‰) = (^aX/^bX)_{sample}/(^aX/^bX)_{standard} - 1, \qquad (1)$$

where $^aX/^bX$ is the ratio between the heavier ($a$; $^{13}C$ or $^{15}N$) and the lighter isotope ($b$; $^{12}C$ or $^{14}N$) of element X (C or N), and the standard is either air for $\delta^{15}N$ or VPDB for $\delta^{13}C$. Nitrogen isotope data were calibrated using IAEA-N-2 ($\delta^{15}N = +20.3 \pm 0.2‰$), IAEA-NO-3 ($\delta^{15}N = +4.7 \pm 0.2‰$) and urea ($\delta^{15}N = +0.3 \pm 0.25‰$), and C-isotope data were calibrated using urea ($\delta^{13}C = -34.1‰$), sucrose ($\delta^{13}C = -10.5‰$), Arndt acetanilide ($\delta^{13}C = -29.5‰$), and an internal laboratory standard (Pennsylvania State University; $\delta^{13}C = -48.3‰$). Nitrogen and TOC abundances were calculated from the integrated $N_2$ and $CO_2$ peak areas, respectively. Replicate analysis of samples and standards demonstrates a relative precision of better than 10%.

The isotopic composition of kerogen-bound nitrogen ($\delta^{15}N_{kerogen}$) was determined in the GAPP Lab (Syracuse University, NY) using an automated Nano-EA system similar to that described by Polissar et al.[72] The use of a Nano-EA system facilitates determination of $\delta^{15}N$ in nitrogen-lean Precambrian kerogens, permitting analysis of 1–3 mg of kerogen equating to 25–100 nanomoles of nitrogen. Sample capsules were evacuated and purged with argon prior to introduction into the EA (Elementar Isotope Cube) to remove interstitial atmospheric nitrogen. A 90-s $O_2$ pulse ensured complete combustion after an initial 45-s He purge. The combustion products were then carried in a He stream (150 ml per min), via oxidation (1100 °C) and reduction (650 °C) reactors, to a silica gel cryotrap immersed in liquid nitrogen. Using a reduced He carrier flow (~2 ml per min), the analyte $N_2$ was introduced to the Isoprime 100 IRMS through an Agilent CarboBond column (25 m × 0.53 mm × 5 µm). Given the high C/N ratios, the $CO_2$ generated during combustion was retained in a molecular sieve trap, eliminating $CO_2$ carryover and preventing the generation of neoformed isobaric CO in the ion source. Additionally, the use of small sample sizes permitted by the Nano-EA technique eliminates the potential for incomplete oxidation and generation of CO during combustion.

Individual kerogen samples were run in triplicate using sequentially larger samples and blank-corrected using Keeling-style plots. International (IAEA N1, +0.4‰; IAEA N2, +20.3‰ and NIST 1547, +2.0‰) and in-house (Messel Oil Shale, +7.0‰) reference materials were run in a similar manner to samples, using equivalent quantities of nitrogen. The resulting blank-corrected sample and standard data were expressed in delta-notation (Eq. 1) and corrected to the certified values of the reference materials using the correction scheme described in Coplen et al.[73] Reproducibility for samples and standards (±0.25‰) approaches that reported for reference materials (±0.2‰) approximating standard EA-IRMS techniques.

**Data availability**. The data that support the findings of this study are available from the corresponding authors upon reasonable request.

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

## Acknowledgements

This work was supported by awards from the Simons Foundation (Simons Collaboration on the Origins of Life) and NSF (EAR-1338810 EAR-1455258). Funding was also provided by the National Key R&D Project of China (2016YFA0601104 and 2016YFC0601005), the 973 program (2013CB955704), the Chinese National Natural Science Foundation (41472170), the 111 project (B08030) and a grant from the National Research Foundation (NRF) in South Africa to N.J.B. The Council for Geoscience in South Africa, specifically those at the National Core Library in Donkerhoek, are thanked for facilitating access to the core materials. Carolyn Colonero is acknowledged for assistance with nitrogen and carbon isotope analyses at MIT. We thank Timothy Lyons for his comments on an earlier version of this manuscript.

## Author contributions

G.L. and R.E.S. designed the research. G.L., S.O. and N.J.B. collected core samples. G.L. and C.K.J. conducted the geochemical analyses. G.L., C.K.J. and G.I. analysed the data and equally wrote the manuscript with insight provided from R.E.S., N.J.B., S.O., T.J.A., Y.C. and S.X.

## Additional information

**Competing interests:** The authors declare no competing interests.

