## [Peer Review File · Nature Communications]

Reviewers' comments:

Reviewer #1 (Remarks to the Author):

The authors present records of C and N isotopes across three cores that have recently been shown to host the disappearance of S-MIF, reflecting oxygenation of the atmosphere. While the dataset could eventually be an important one, its current interpretation is flawed and overly complicated. The data are interpreted based on the assumption that an anoxic atmosphere (i.e., occurrence of S-MIF) necessitates a fully anoxic ocean, which is not a valid assumption. A simple comparison of the d15N data presented in this study with modern-to-recent d15N records would suggest that the dataset in its entirety could be explained by a modern-style aerobic N cycle controlled by the balance of N fixation and denitrification/anammox. The occurrence of N fixation is independent of redox, and the occurrence of nitrification/denitrification would only require localized oxygen in the oceans, which arguably predated the rise of O₂ in the atmosphere. The partial nitrification scenario presented here is unconvincing and unnecessary.

In addition to problems with data interpretation, there are several other issues that warrant significant revision before publication:

1) The manuscript is missing a rigorous evaluation of post-depositional alteration and its possible effects on the bulk d15N signals. In particular, the role of isotope exchange during post-depositional fluid flow and the possibility of significant input of terrigenous material should both be addressed and ruled out before the d15N data can be reliably interpreted as reflecting the marine N cycle.

2) The manuscript is also missing any discussion of the cores, their facies or depositional context, referring the reader to a previous paper for this information. Given that the N cycle is incredibly spatially heterogeneous even in the modern oceans, it is imperative that the depositional environment be discussed and considered within this manuscript, such that readers do not have to look elsewhere for this critical context.

3) The manuscript is missing some key references (e.g., updated age dates in Rasmussen et al., 2013, EPSL), and fails to acknowledge important previous work done on this section, notably the d13Corg data and interpretations of Coetzee et al., 2006, Journal of South African Geology.

4) Figure 1 includes d34S and D33S data that are not discussed or referred to in the text. Presumably the D33S is shown to demonstrate the loss of S-MIF, but there are also large fluctuations in the d34S data that are not explained – if this data is included in this manuscript the reader should not have to go to another paper for its discussion or interpretation.

The authors should also be aware that another paper is currently in press at Nature that examines high-resolution d15N and d13C records across the Rooihogte-Timeball Hill formation boundary alongside Fe speciation data as a proxy for water column redox. They will need to carefully consider the findings of that work when revising the interpretations of their data.

Some specific comments related to those above:

Lines 54-57, Line 64: Here and throughout, it is important to note that the aerobic N cycle requires oxygen in the water column/local environment, independent of pO₂ concentrations in the atmosphere.

Lines 71-76: In fact, significant nitrogen isotope exchange can occur between mineral N (e.g., ammonium hosted in clays) and N-containing compounds during metasomatism or when fluids migrate during organic matter maturation, producing fairly large offsets between d15Norg and d15Nbulk of > 5‰. In addition, shallow water sediments such as these (which were deposited in a

deltaic environment that could have had significant influence from riverine sources) could have addition of older, terrigenous material washed in from land. Thus bulk rock $\delta^{15}\text{N}$ values cannot be assumed to be representative of marine organic matter without discounting these possibilities.

Lines 81-83: This is insufficient description of the studied section. There is no discussion of the facies or depositional environment(s) represented by these cores. Given that the N cycle is extremely spatially heterogeneous even in modern environments, depositional context is critical to determine the validity of the interpretation of the data, and should be explicitly discussed within this paper. Also, as the Rooihogte and Timeball Hill Formations are interpreted as being deposited on a paleo-delta slope (see, e.g., Coetzee et al., 2006), this has obvious implications for potential detrital N sources (as above).

Line 89: Also cite Bekker et al., 2004, *Nature*, who first noted the disappearance of S-MIF in this section.

Lines 106-onward: The authors should acknowledge and consider the original work by Coetzee et al., 2006, *South African Journal of Geology*, which first measured $\delta^{13}\text{C}_{\text{org}}$ in the Rooihogte and Timeball Hill Formations, and interpreted these values within the context of the biogeochemical carbon cycle and global redox evolution.

Figure 1. It is unclear why sulfur isotope data is included in this figure. These data are not referred to or discussed anywhere in this manuscript. The loss of S-MIF is mentioned, but there are also large changes in $\delta^{34}\text{S}$ that are presumably caused by changes in redox/biogeochemical S cycling that are not discussed in the manuscript. Also, please include updated ages of Rasmussen et al., 2013, *EPSL*.

Lines 122-126: This brief discussion and C/N/isotope crossplots are insufficient to rule out post-depositional alteration, particularly via exchange with metamorphic fluids, or terrigenous input of NH_4^+ -bearing clays – see above. In order to interpret the $\delta^{15}\text{N}_{\text{bulk}}$ data as representative of marine N cycling, the authors need to compare $\delta^{15}\text{N}_{\text{bulk}}$ values with $\delta^{15}\text{N}_{\text{org}}$ values from the same section to demonstrate that these values are representative, and/or they should present additional supporting data to illustrate this point, e.g., crossplots of TN and $\delta^{15}\text{N}$ versus wt% K₂O to demonstrate that fluid exchange during K-metasomatism has not significantly altered the bulk $\delta^{15}\text{N}$ values; plots of TN/ $\delta^{15}\text{N}$ versus something like Al or Ti to demonstrate that the N is not detrital in origin.

Lines 132-140: Again, see comments above. Post-depositional isotope exchange can sometimes result in a correlation between Corg/N and $\delta^{15}\text{N}_{\text{bulk}}$, but a lack of this correlation is insufficient to rule out post-depositional alteration by this process. Additional data is required.

Lines 148-154: The discussion of the Neoproterozoic $\delta^{15}\text{N}$ records is incomplete and somewhat misleading. The Garvin and G&F studies showed 2-5‰ increases in $\delta^{15}\text{N}$ to near modern values (5-7‰, not what I would consider “strong positive values”) within a single stratigraphic interval of deepwater sediments. These changes were interpreted to reflect only transient or localized aerobic N cycling, and more recent work (Busigny et al., 2013, *Chemical Geology*) has suggested that these deep water signals could also be caused by N redox cycling independent of surface oxygenation (e.g., involving Fe-oxides).

Lines 160-162: Again, aerobic N cycling would occur independently from atmospheric O_2 , as it only requires localized O_2 in the oceans.

Line 167: Similarly, S-MIF does not require a fully anoxic ocean, it just requires an anoxic atmosphere. The authors have no evidence that an anoxic ocean persisted during deposition of the Carltonville sequence, but they make this assumption throughout the paper and base their interpretations of the stable isotope records on it. Given the number of recent papers suggesting

localized marine O₂ as far back as ~3.0 Ga, this was likely not the case (see also Zerkle et al., Nature, in press).

Line 176-178: The statements that “the majority of d¹⁵N values are significantly higher than those of Phanerozoic counterparts”, and that the N cycle prior to the GOE was thus “radically different” from that of the Phanerozoic and modern oceans are fundamentally incorrect. In fact the majority of the d¹⁵N values the authors present here (which vary from ~6-8‰) are identical to the mean d¹⁵N values of modern marine organic matter (~6-7‰; Peters et al., 1978, L&O). Even higher sedimentary d¹⁵N values can be generated in modern upwelling zones associated with intensive denitrification (up to 15‰; e.g., De Pol-Holz et al., 2009, Deep Sea Research). Therefore, the data presented here are fully consistent with a modern-style aerobic N cycle, with d¹⁵N values representing the balance between the input of N into the marine system as N₂ fixation in open ocean settings and the loss of fixed N via denitrification/anammox in anaerobic environments/oxygen minimum zones.

Lines 180-215: The incomplete nitrification scenario is unlikely and inconsistent with the data. This idea was first invoked by Thomazo et al. to explain extremely ¹⁵N-enriched values (> +50‰) alongside extremely ¹³C-depleted values (< -50‰) in the Tumbiana Formation. However, the assumed fractionation is based on a single set of experiments (Mandernack et al., 2009, Chemical Geology) during N₂O production via co-oxidation of ammonium by methanotrophs under laboratory conditions – this process has only been shown to occur in terrestrial wetlands and soils. It is not a well-known or well-established part of the modern marine N cycle, and similar isotope effects have never been documented anywhere else in the rock record or in any modern marine system (even in low-O₂ areas). The more recent hypothesis of Stueken et al., that the Tumbiana was a high-pH alkaline lake and the extreme d¹⁵N values could be produced by degassing of ammonia is more likely. Regardless, this scenario does not explain the isotope records presented in this paper and, as above, these records can quite easily be explained by modern aerobic N cycling processes if a fully anoxic ocean is not assumed.

Lines 220-231: I agree that d¹⁵N values of ~2‰ suggest an enhanced contribution from N fixation, but this is independent of ocean (or atmospheric) redox. Phanerozoic OAE's generally have much lower d¹⁵N values (down to -5‰; Jenkyns et al., 2007, 2001; Rau et al., 1987; Higgins et al., 2012; Kuypers et al., 2004; etc), interpreted as reflecting: 1) enhanced N fixation due to P release from sediments under low O₂ bottom water conditions; 2) larger fractionations from cyanobacteria fixing N at the high [Fe] typical of low-O₂ waters; and/or 3) ammonium uptake. More typical d¹⁵N values of ~2‰ are characteristic of modern aerobic marine settings dominated by N₂ fixation, such as in open ocean settings where nutrient upwelling is minimal, and do not require an anoxic water column or enhanced ammonium availability (in fact they argue against the latter).

Line 223: What do you mean by a nitrogen cycle “typical of modern marine anoxic environments”? Anaerobic N cycling in modern ocean sediments and OMZ's is dominated by denitrification and anammox, the isotopic fingerprint of which is evident throughout the entirety of this section, and is the opposite of what you see here. As above, I agree that d¹⁵N values of +2‰ suggest that nitrogen fixation was the dominant process in this part of the section, but that could be due to high levels of primary productivity, high P input via weathering, and/or even a change in depositional environment, none of which are redox dependent.

Reviewer #2 (Remarks to the Author):

I spent a lot of time thinking about this paper. The results are curious. One of the strongest geochemical signals that has come to define the GOE is the onset of the mass dependent isotopic fractionation of S. That signal has been interpreted as the oxygen dependent production of

stratospheric ozone. The build up of oxygen in the atmosphere had to have altered the N cycle – and the data presented here clearly show a positive isotopic fractionation in the three cores. However, unfortunately, the data don't show that the signal either rose from a lower value deeper in time, or that it underwent fluctuations – except after the GOE. It is hard to reconcile oxygen “oases” when the atmosphere was showing signs of ozone. The authors clearly understand the fundamental issue, and spend a significant amount of effort to (1) justify why they didn't measure the N isotopes specifically in a kerogen fraction (i.e., that the results are not an artifact), and (2) account for the lower isotopic fractionation after the GOE. Obviously it is very hard to justify the use of bulk N isotopes without even calibrating the data against a couple measurements made on kerogens – but I will assume for argument's sake that the measurements are as valid as can be expected. That said, what is not clear is how the rise of N could not have led to denitrification and a reduction in carbon burial. The authors should read (or reread) the Fennell et al paper ([https://marine.rutgers.edu/~kfennel/Fennel%20et%20al%202005%20AJ S.pdf](https://marine.rutgers.edu/~kfennel/Fennel%20et%20al%202005%20AJ%20S.pdf)). If the model presented by Fennell et al., is wrong, what happened that led to the GOE? If the model is correct, then why do we start out at such high ^{15}N values in the three cores?

Reviewer #3 (Remarks to the Author):

Review of the paper entitled: Onset of modern-style nitrogen cycle following initial buildup of atmospheric oxygen

Written by Genming Luo et al.

This study mostly focuses on deciphering the evolution of the nitrogen biogeochemical cycling across the GOE using nitrogen isotopes analyses of bulk rocks and organic matter carbon isotopes. These results are compared to previous interpretations placing the oxygenation of Earth atmosphere following the disappearance of MIF of sulfur isotopes within the studied sections. Using isotopic signatures the authors developed a multi step model of evolution of the nitrogen biogeochemical cycle from ammonium oxidation in oxygen oases to complete denitrification and finally modern style cycle 70 Ma after the GOE. The carbon isotopes variations are interpreted to underline aerobic methanotrophic metabolism coupled to ammonium oxidation as previously proposed in the literature for older rocks deposit (ca. 2.7 Ga).

The novelties of this study lie on the first record of nitrogen isotopes across a well-studied sedimentary deposit encompassing the GOE (based on MIFS).

The paper is well written, illustrations however could/might be improved. Comparison with previous studies and bibliography are well done and underline the high potential of nitrogen isotopes to bring further insights on the “oxygenation” and life history on Earth. The reading of this manuscript raises one major question that in my opinion need to be address and more deeply discuss.

It concerns the C/N ratio reported in Fig.3 and data tables.

I do agree that thermal metamorphism is expressed in the reported rock record and that high C/N ratio reflects thermal devolatilization of N with little isotope change (lines 128-132). However the low (below Redfield!) C/N needs to be explained a little bit more. If it does indeed reflect adsorption on phyllosilicate it should be reflected in the lithology and a relationship between potassium content and C/N ratios ($\delta^{15}\text{N}$ to testify as well) for instance is expected. If some samples show a stronger nitrogen removal than carbon and the others the way around they have to be discussed and testify for secondary alteration of their isotope signal independently. Since speciation is also different the reader is expecting a more robust demonstration that secondary processes does not impact the record (4 to 5 permille shift is in the range of expected secondary effects of combined devolatilization in one hand and remineralization on clay on the other hand). At least I suggest sorting the data of Fig.3B with respect to lithology and possible nitrogen speciation to discuss secondary processes.

One example of this form table data:

KEA-4 551.04-551.13-B 551,08 8,8 1388,2 -32,6 0,43 3,6

KEA-4 551.04-551.13-A 551,08 7,5 645,5 -35,0 6,94 125,4

If I understand well it is slabs of the same sample core, the C/N difference is very high (and so speciation probably), the d15N also quite variable 1.3 permille (note by the way that point is needed not coma).

How do you explain this? Did you analyze a piece of mica in the sample A? This need to be clarify I guess and replace in a context of what the data reflect in terms of primary and secondary processes.

This said, I agree with the conclusions even if some stay rather theoretical. I highly recommend to depict suggest changes in the biogeochemical cycling of nitrogen through time by adding a step cartoon representing changes in speciation and reactions.

This is overall a good and interesting contribution for Nat.com journal, which will find easily a broad audience after some clarification.

Some suggestions in order to further improve the manuscript are listed bellow:

Line 24: high-resolution is overtake

Line 38: not clear what means "substantive oxidative nitrogen cycle" you and others described oxidation reactions well before the GOE (cf. line 159) and the reaction you suggest after the GOE is characterizing the cycle in "modern anoxic environment". A cycle where nitrification could be quantitative may be but I suggest to stay on that bottom line to avoid misunderstanding.

Line 69: may cite here Ader et al., 2016 as it is a review paper on that question

Line 76: again you may refer to Ader et al., 2016 that compile d15N bulk-ker offset for Precambrian sediments.

Line 165: at least a ref for this inference

Line 193: precise age of the Tumbiana Fm.

Line 210: argues against incomplete turn to - argue for complete

Line 219: ref for the "typical" d13Corg values (and a delta is missing)

Line 209: The question of the mass balance is interesting but also the ammonia oxidizing bacteria and methanotrophs able to co-oxidize ammonia and methane have different fractionation (around 20 compare to 55 permille, respectively) ...

Line 227: In your step scenario there is never a biogeochemical cycling allowing for a growing nitrate oceanic reservoir because denitrification is always quantitative – so "modern style nitrogen cycle" is quite overtake

Line 239: once again oxidative is not appropriate I think, after all there is no evidence for a substantial ox-nitrogen species reservoir in the whole dataset.

C.Thomazo

RESPONSES TO THE REVIEWERS COMMENTS:

Buoyed by the reviewers' comments, we have decided to redraft our manuscript for reconsideration for publication in *Nature Communications*. We received three sets of reviewers' comments, which we have considered and implemented while redrafting this resubmission. Here we detail our responses to those comments on a point-by-point basis organised by specific reviewers. Their original text is provided first in orange followed by our responses in black. Given the substantial overhaul, our newly submitted manuscript bears little resemblance to the initial submission and therefore line references given by the reviewers do not easily translate.

REVIEWER 1:

Reviewer 1 provided an interesting and alternative view point. Largely we do not agree with their interpretation of our data yet we welcome their objection because it has encouraged us to think more critically while interrogating our dataset, the current understanding of the nitrogen cycle and how it would have evolved in a planetary sense. R1 structures their review as a series of major points followed by other issues that they believed warranted revision before publication. Here we deal with those in turn.

Overview and Major Concerns

The authors present records of C and N isotopes across three cores that have recently been shown to host the disappearance of S-MIF, reflecting oxygenation of the atmosphere. While the dataset could eventually be an important one, its current interpretation is flawed and overly complicated. The data are interpreted based on the assumption that an anoxic atmosphere (i.e., occurrence of S-MIF) necessitates a fully anoxic ocean, which is not a valid assumption. A simple comparison of the $\delta^{15}\text{N}$ data presented in this study with modern-to-recent $\delta^{15}\text{N}$ records would suggest that the dataset in its entirety could be explained by a modern-style aerobic N cycle controlled by the balance of N fixation and denitrification/anammox. The occurrence of N fixation is independent of redox, and the occurrence of nitrification/denitrification would only require localized oxygen in the oceans, which arguably predated the rise of O_2 in the atmosphere. The partial nitrification scenario presented here is unconvincing and unnecessary.

We thank R1 for their encouragement and apologise for the confusion. We do not believe that an essentially oxygen-free atmosphere translates directly into an anoxic ocean. In fact, to consume reductants and oxygenate the atmosphere the oceans must have been a net source of oxygen, and other gases, to the atmosphere (e.g., Izon et al., 2017). That said, it is widely believed that the persistence of mass-independent S-isotope fractionation requires the atmosphere to have had vanishingly low free-oxygen levels (10^{-5} – 10^{-7} present atmospheric levels, e.g., Claire et al., 2014). Therefore, it is not unreasonable to assume that unless the site in question was a site of active oxygen production then the dissolved oxygen content would have also been exceedingly low. The persistence of dissolved oxygen requires that its rate of production outpaces its destruction by a reducing atmosphere from above and a ferruginous ocean at depth. This is supported by modelling and geochemical constraints that indicate dissolved oxygen contents probably didn't exceed a couple of micromoles (Waldbauer et al., 2011; Hardisty et al., 2014; Kipp et al., 2017). Contrary to R1's opinion, we believe that this led to the operation of nitrogen cycle that only superficially reflects its contemporary counterpart and the similarities between modern and Palaeoproterozoic $\delta^{15}\text{N}$ values are better explained as part of an evolving nitrogen cycle.

These arguments form the centrepiece of our manuscript:

- 1) Nitrate would have been quantitatively lost via denitrification in O_2 -limited and stratified oceans. Nitrate and nitrite are important electron acceptors, which directly follow O_2 in the oxidation-reduction series. Therefore, both oxidised nitrogen-species would have been quantitatively reduced to N_2 either through denitrification or anammox before and in the immediate aftermath of the GOE, thus, preventing their accumulation. This is consistent with modelling results that suggest denitrification rate would have exceeded nitrification

rate when dissolved O₂ content was less than 11 μM (Fennel et al., 2005), as was the case at and around the GOE. Therefore, we seek other mechanisms to explain the positive δ¹⁵N_{kerogen} values rather than concluding a modern nitrogen cycle was established.

2) There is a large N-isotope fractionation associated with nitrification, which preferentially leads to the formation of ¹⁵N-depleted nitrate and ¹⁵N-enriched ammonium. Loss of ¹⁵N-depleted nitrate as dinitrogen gas during denitrification creates a ¹⁵N-enriched residual ammonium pool, leading to the positive δ¹⁵N values recorded in sediments. Therefore, we agree with R1 that nitrification required only localized oxygen in the ocean, and that it developed before the disappearance of S-MIF. The ammonium oxidation process requires strong oxidants such as O₂. As we discussed above, the O₂ content in the surface ocean was limited, and thus we suggest that the ammonium oxidation would have been incomplete. This is a subtle but significant difference from the contemporary nitrogen cycle.

In summary, we argue that the Palaeoproterozoic marine nitrogen cycle was characterised by a lower nitrification rate relative to that of denitrification and/or anammox and ammonium was the main nitrogen species in the ocean. These characteristics are different from those operating in the modern ocean, which features quantitative nitrification and only partial denitrification within restricted OMZs. After planetary oxygenation, continued accumulation of atmospheric O₂ may have facilitated complete nitrification and denitrification/anammox, prompting the diazotrophy that we see at the top of the examined succession.

Minor Concerns and Comments:

R1.1: The manuscript is missing a rigorous evaluation of post-depositional alteration and its possible effects on the bulk δ¹⁵N signals. In particular, the role of isotope exchange during post-depositional fluid flow and the possibility of significant input of terrigenous material should both be addressed and ruled out before the δ¹⁵N data can be reliably interpreted as reflecting the marine N cycle.

In response to R1's concerns, we generated a complementary set of δ¹⁵N data from kerogen isolates. Our new kerogen data (δ¹⁵N_{kerogen}) show similar trends to those observed initially in our bulk records, with a small offset. The similar trends preserved in both δ¹⁵N datasets, the large variability they record, and the more-or-less constant offset between these profiles suggests that the primary N-isotopic compositions of the study samples have not been significantly and differentially overprinted by diagenesis. The small δ¹⁵N offsets of kerogen N versus bulk N (~2–4 ‰) that are present in our datasets are apparently typical of marine sediments and indeed mimic those seen in core EBA-2 reported by Zerkle et al. (2017).

The terrigenous nitrogen flux would have presumably been very small on the early Earth owing to a limited terrestrial biosphere. Accordingly, the potential influence from detrital N would have been very small, as concluded by Zerkle et al. (2017) via examination of K and N relationships in core EBA-2. Although we cannot isolate such effects on the nitrogen isotope compositions of our samples, the parallels between the bulk and kerogen δ¹⁵N data, and the lack of any evidence for differential fluid alteration, are solid evidence that near-primary δ¹⁵N values have been preserved.

Following R1's advice, however, we have added a discussion and an additional figure (Figure 4) that details these arguments in the section entitled "*The Fidelity of the Nitrogen Isotope Records*". To further proceed with caution, our discussion centres on the kerogen-sourced δ¹⁵N data.

R1.2. The manuscript is also missing any discussion of the cores, their facies or depositional context, referring the reader to a previous paper for this information. Given that the N cycle is incredibly spatially heterogeneous even in the modern oceans, it is imperative that the depositional environment be discussed and considered within this manuscript, such that readers do not have to look elsewhere for this critical context. We apologise for this oversight. We were trying to keep the manuscript succinct and focused. In the redrafted submission, we have added details of the individual cores and depositional context in the revised manuscript (Line 106-125). This is also clarified by the addition of Figure 1.

R1.3. The manuscript is missing some key references (e.g., updated age dates in Rasmussen et al., 2013, EPSL), and fails to acknowledge important previous work done on this section, notably the $\delta^{13}\text{C}_{\text{org}}$ data and interpretations of Coetzee et al., 2006, *Journal of South African Geology*. Again, we apologise for this oversight and thank R1 for pointing this out. All the missing references have been added as requested. See lines 112 and 115 for examples.

R1.4. Figure 1 includes $\delta^{34}\text{S}$ and $\Delta^{33}\text{S}$ data that are not discussed or referred to in the text. Presumably the $\Delta^{33}\text{S}$ is shown to demonstrate the loss of S-MIF, but there are also large fluctuations in the $\delta^{34}\text{S}$ data that are not explained – if this data is included in this manuscript the reader should not have to go to another paper for its discussion or interpretation. We agree, the S-isotope data are peripheral to the nitrogen-story and have been published recently. Rather than complicate the manuscript with extraneous data, we have removed these data from Figure 1. We now refer to the intervals as defined in Luo et al. (2016).

R1.5. The authors should also be aware that another paper is currently in press at *Nature* that examines high-resolution $\delta^{15}\text{N}$ and $\delta^{13}\text{C}$ records across the Rooihogte–Timeball Hill formation boundary alongside Fe speciation data as a proxy for water column redox. They will need to carefully consider the findings of that work when revising the interpretations of their data. While it is a pity that Zerkle and colleagues' manuscript was accepted for publication in *Nature* while ours was in review, as noted by Reviewer 1, N-isotope systematics are inherently locally-driven and therefore the conclusions drawn in the Zerkle et al. study require to be tested. Our study differs from and represents an advance over the Zerkle *et al.* (2017) study for multiple reasons:

1. In addition to core EBA-2, we generated data for two additional cores (KEA-4 and EBA-4) extending the spatial resolution across the larger Transvaal Basin.
2. All our data have been generated from sample splits that have previously been the subject of quadruple S-isotope analysis (Luo et al., 2016). The S-isotope data allow us to directly constrain the GOE, and how the nitrogen cycle responded to planetary oxygenation without troublesome and imprecise correlations. Moreover, using this S-isotope chemostratigraphic framework, we have generated 19 kerogen $\delta^{15}\text{N}$ data points in the S-MIF and transitional interval, versus only one kerogen-sourced data point in Zerkle *et al.*, (2017), which more completely reveals the pre-GOE nitrogen cycle that was under sampled in previous work.
3. Finally, we extended our attention to the top of the Timeball Hill Formation in all three study cores. This contrasts with previous work, which focuses on the younger Rooihogte Formation up into the lower Timeball Hill Formations. Absolute dating suggests that the uppermost Timeball Hill Formation might record the prelude to the oxygen overshoot during the Lomagundi event (e.g., Bekker et al., 2012). Therefore, our work covers a larger spectrum of atmospheric $p\text{O}_2$ than in Zerkle et al. (2017) which only focuses on the immediate aftermath of the GOE. These data are unique, and time-equivalent points have not been published elsewhere. These data form one of the main findings of our new submission, and capture the response of the nitrogen cycle to rising $p\text{O}_2$.

In summary, we have tried to contextualise our data considering a paper that was not published at the time we initially submitted our own. We have made every effort to credit Zerkle et al. (2017) and be clear where our study complements, contrasts, or advances that work. We present new ideas, but hopefully in a more balanced approach, that challenges how the nitrogen cycle evolved.

Specific Comments:

Lines 54–57, Line 64: Here and throughout, it is important to note that the aerobic N cycle requires oxygen in the water column/local environment, independent of $p\text{O}_2$ concentrations in the atmosphere. We agree. See our previous response to R1's general objection.

Lines 71-76: In fact, significant nitrogen isotope exchange can occur between mineral N (e.g., ammonium hosted in clays) and N-containing compounds during metasomatism or when fluids migrate during organic matter maturation, producing large offsets between $\delta^{15}\text{N}_{\text{Org}}$ and $\delta^{15}\text{N}_{\text{Bulk}}$ of >

5 ‰. In addition, shallow water sediments such as these (which were deposited in a deltaic environment that could have had significant influence from riverine sources) could have addition of older, terrigenous material washed in from land. Thus, $\delta^{15}\text{N}_{\text{Bulk}}$ values cannot be assumed to be representative of marine organic matter without discounting these possibilities. We agree. See response R1.1.

Lines 81-83: This is insufficient description of the studied section. There is no discussion of the facies or depositional environment(s) represented by these cores. Given that the N cycle is extremely spatially heterogeneous even in modern environments, depositional context is critical to determine the validity of the interpretation of the data, and should be explicitly discussed within this paper. Also, as the Rooihogte and Timeball Hill Formations are interpreted as being deposited on a paleo-delta slope (see, e.g., Coetzee et al., 2006), this has obvious implications for potential detrital N sources (as above). We agree, in fact the heterogeneity in the nitrogen cycle is what initially motivated the multi-core approach. Consequently, our data, and the similar trends observed in all the cores, fortify our conclusions. As mentioned previously, we have added discussion of potential post-depositional alteration and added a kerogen-derived dataset to address any terrestrial contributions. See response to R1.2.

Line 89: Also, cite Bekker et al., 2004, *Nature*, who first noted the disappearance of S-MIF in this section. Bekker et al. (2004), has been added as requested. However, as R1 is aware, this work did not directly note the disappearance of S-MIF, it was an inference. The Bekker et al. study noted the oldest known occurrence of S-MDF. We had to change the wording to accommodate this reference.

Lines 106-onward: The authors should acknowledge and consider the original work by Coetzee et al., 2006, *South African Journal of Geology*, which first measured $\delta^{13}\text{C}_{\text{Org}}$ in the Rooihogte and Timeball Hill Formations, and interpreted these values within the context of the biogeochemical carbon cycle and global redox evolution. As noted in the recent paper by Zerkle et al. (2017), the $\delta^{13}\text{C}_{\text{Org}}$ values in cores EBA-2 and KEA-4 are relatively stable (Figures 2–3). As the carbon isotope data were not the focus of the present study, we did not see a need to elaborate on them. They are described in the results section and mentioned briefly in passing. The reference, however, has been added to the manuscript.

Figure 1. It is unclear why sulfur isotope data is included in this figure. These data are not referred to or discussed anywhere in this manuscript. The loss of S-MIF is mentioned, but there are also large changes in $\delta^{34}\text{S}$ that are presumably caused by changes in redox/biogeochemical S cycling that are not discussed in the manuscript. Also, please include updated ages of Rasmussen et al., 2013, *EPSL*. As before, the sulfur isotope data were removed from Figure 1 and the zonal scheme as described by Luo et al. (2016) was added. The ages reported by Rasmussen et al. (2013) were also added. See response to R1.4.

Lines 122-126: This brief discussion and C/N/isotope cross-plots are insufficient to rule out post-depositional alteration, particularly via exchange with metamorphic fluids, or terrigenous input of NH_4^+ -bearing clays—see above. To interpret the $\delta^{15}\text{N}_{\text{Bulk}}$ data as representative of marine N cycling, the authors need to compare $\delta^{15}\text{N}_{\text{Bulk}}$ values with $\delta^{15}\text{N}_{\text{Org}}$ values from the same section to demonstrate that these values are representative, and/or they should present additional supporting data to illustrate this point, e.g., cross-plots of TN and $\delta^{15}\text{N}$ versus Wt %. K_2O to demonstrate that fluid exchange during K-metasomatism has not significantly altered the bulk $\delta^{15}\text{N}$ values; plots of TN/ $\delta^{15}\text{N}$ versus something like Al or Ti to demonstrate that the N is not detrital in origin. This is repetition of a previous point. Please refer to response R1.1. Given Zerkle et al. (2017) presented potassium data from core EBA-2 and concluded there was limited fluid-driven nitrogen exchange. We opted to generate a more comprehensive kerogen-derived dataset to minimize the effects listed above.

Lines 132-140: Again, see comments above. Post-depositional isotope exchange can sometimes result in a correlation between $\text{C}_{\text{Org}}/\text{N}$ and $\delta^{15}\text{N}_{\text{Bulk}}$, but a lack of this correlation is insufficient to

rule out post-depositional alteration by this process. Additional data is required. This is repetition of a previous point. Please refer to response R1.1.

Lines 148-154: The discussion of the Neoproterozoic $\delta^{15}\text{N}$ records are incomplete and somewhat misleading. The Garvin and G&F studies showed 2–5 ‰ increases in $\delta^{15}\text{N}$ to near modern values (5–7 ‰, not what I would consider “strong positive values”) within a single stratigraphic interval of deep-water sediments. These changes were interpreted to reflect only transient or localized aerobic N cycling, and more recent work (Busigny et al., 2013, *Chemical Geology*) has suggested that these deep-water signals could also be caused by N redox cycling independent of surface oxygenation (e.g., involving Fe-oxides). We apologise for the confusion; this text has been removed along with subjective language throughout the new manuscript. In our defence, the values we quoted are distinctly larger than those produced by anaerobic nitrogen cycling, which can only generate muted signals. We agree that the data reported in Garvin and G&F reflect aerobic N cycling processes generating and destroying nitrate. Whether it was transient, however, is another question. The samples analysed here were deposited in shallow-water environments and are depleted in iron oxides (see Fe-speciation in Zerkle et al., 2017). Thus, ammonium oxidation by Fe-oxides would have been unlikely.

Lines 160-162: Again, aerobic N cycling would occur independently from atmospheric O_2 , as it only requires localized O_2 in the oceans. As before, see our response to the general comment above.

Line 167: Similarly, S-MIF does not require a fully anoxic ocean, it just requires an anoxic atmosphere. The authors have no evidence that an anoxic ocean persisted during deposition of the Carletonville sequence, but they make this assumption throughout the paper and base their interpretations of the stable isotope records on it. Given the number of recent papers suggesting localized marine O_2 as far back as ~3.0 Ga, this was likely not the case (see also Zerkle et al., *Nature*, in press). As before, we do not argue for an anoxic ocean, and we agree with R1 that localised marine O_2 was present during the S-MIF intervals. We also would like to reiterate that given the short atmospheric residence time of O_2 in a pre-GOE atmosphere then O_2 must be generated *in situ*—It cannot be transported. I think we all agree this is in stark contrast to our contemporary planet and has the potential to influence the nitrogen cycle. The Fe-speciation data (Zerkle et al., 2017) also speak to vanishingly low oxygen concentrations.

Line 176-178: The statements that “the majority of $\delta^{15}\text{N}$ values are significantly higher than those of Phanerozoic counterparts”, and that the N cycle prior to the GOE was thus “radically different” from that of the Phanerozoic and modern oceans are fundamentally incorrect. In fact the majority of the $\delta^{15}\text{N}$ values the authors present here (which vary from ~6–8‰) are identical to the mean $\delta^{15}\text{N}$ values of modern marine organic matter (~6–7‰; Peters et al., 1978, L&O). Even higher sedimentary $\delta^{15}\text{N}$ values can be generated in modern upwelling zones associated with intensive denitrification (up to 15‰; e.g., De Pol-Holz et al., 2009, *Deep Sea Research*). Therefore, the data presented here are fully consistent with a modern-style aerobic N cycle, with $\delta^{15}\text{N}$ values representing the balance between the input of N into the marine system as N_2 fixation in open ocean settings and the loss of fixed N via denitrification/anammox in anaerobic environments/oxygen minimum zones. We apologise for this confusion. When we previously referred to ‘Phanerozoic counterparts’ we meant the episodes where anoxic environments were significantly expanded, such as the oceanic anoxic events (OAEs) in Cretaceous. Explicitly, we do not wish to make the inferred comparison between the Palaeoproterozoic ocean and its contemporary analogue, given that the former was almost completely devoid of oxygen and the latter the opposite. This subjective language has been removed from the manuscript throughout.

Lines 180-215: The incomplete nitrification scenario is unlikely and inconsistent with the data. This idea was first invoked by Thomazo et al. to explain extremely ^{15}N -enriched values (> +50‰) alongside extremely ^{13}C -depleted values (<-50‰) in the Tumbiana Formation. However, the assumed fractionation is based on a single set of experiments (Mandernack et al., 2009, *Chemical Geology*) during N_2O production via co-oxidation of ammonium by methanotrophs under laboratory

conditions—this process has only been shown to occur in terrestrial wetlands and soils. It is not a well-known or well-established part of the modern marine N cycle, and similar isotope effects have never been documented anywhere else in the rock record or in any modern marine system (even in low-O₂ areas). The more recent hypothesis of Stueken et al., that the Tumbiana was a high-pH alkaline lake and the extreme $\delta^{15}\text{N}$ values could be produced by degassing of ammonia is more likely. Regardless, this scenario does not explain the isotope records presented in this paper and, as above, these records can quite easily be explained by modern aerobic N cycling processes if a fully anoxic ocean is not assumed. While we thank R1 for their opinion (that our model is unlikely), they are categorically wrong to state that our model is inconsistent with the data. We agree the positive $\delta^{15}\text{N}$ data at/around the GOE can be explained by partial denitrification, as discussed in their recent paper. We disagree, however, with their view that this is the only explanation of the data and we argue that incomplete nitrification is a more satisfactory explanation of the data. As we detailed previously, we did not assume that the pre-GOE ocean was fully anoxic. That said, the persistence of S-MIF and deposition of BIFs demonstrates that both the atmosphere and deep ocean were anoxic, which limits the development of oxygenated water masses to sites of oxygen production. While the Fe-speciation data presented by Zerkle et al. (2017) suggest that EBA-2 was a periodically a site of oxygen production ($\text{Fe}_{\text{HR}}/\text{Fe}_{\text{T}} < 0.22$), most their data are either equivocal ($\text{Fe}_{\text{HR}}/\text{Fe}_{\text{T}} 0.22\text{--}0.38$) or suggest anoxic conditions ($\text{Fe}_{\text{HR}}/\text{Fe}_{\text{T}} > 0.38$)—consistent with their interpretation in proximity to a dynamic chemocline. Clearly, as demonstrated by geochemical proxies (e.g., I/Ca records (Hardisty et al., 2014) and selenium isotopes (Kipp et al., 2017)) and modelling results (Waldbauer et al., 2011), the oxygen concentrations were very low (less than a couple of micromoles) during this interval. Modelling studies argue that denitrification would have exceeded nitrification in an oxygen-limited environment ($< 11 \mu\text{M}$). Accepting near-quantitative denitrification, the isotopic effect invoked to explain modern positive $\delta^{15}\text{N}$ values would have been nullified. Thus, necessitating an alternate explanation for the high $\delta^{15}\text{N}$ values seen in our pre- and early GOE samples: partial ammonium oxidation (nitrification). Here, ^{14}N is preferentially transferred to nitrate and lost as $^{14}\text{N}_2$, resulting in a residual pool of $^{15}\text{NH}_4^+$ that is assimilated and preserved as in the sediments with positive $\delta^{15}\text{N}$.

In our manuscript, as suggested by R1, we credit Thomazo et al. (2011) with conceiving the idea that partial denitrification may result in positive $\delta^{15}\text{N}$. However, contrary to the argument presented above by R1, there are significant differences between our dataset and that from the Tumbiana Fm presented by Thomazo et al. (2011) — significantly lower $\delta^{15}\text{N}$ and higher $\delta^{13}\text{C}$. As we outline in our manuscript we suggest that co-oxidation of ammonium by methanotrophs (Thomazo et al. 2011) was not important in the succession studied here because the $\delta^{13}\text{C}_{\text{Org}}$ data do not descend to the necessary low values indicative of methanotrophy and no significant correlation is present between $\delta^{13}\text{C}_{\text{Org}}$ and $\delta^{15}\text{N}_{\text{kerogen}}$. Instead we call on partial ammonium oxidation by ammonium-oxidizing bacteria (AOB) or ammonium-oxidizing archaea (AOA). Multiple studies, have shown that incomplete nitrification enriches ^{15}N in the residual ammonium (Mariotti et al., 1981; Casciotti et al., 2003; Santoro and Casciotti, 2011) and could have produced the positive $\delta^{15}\text{N}$ values we see. Taken further, we agree, that the experiments that underpin the Thomazo et al. study are not relevant here. However, the recent hypothesised alternative explanation by Stueken et al., (2015), presented by R1, is also far from being perfect, and is reliant on the Tumbiana formation representing a lacustrine environment. This is contested by some, although the balancing argument is not clear in that paper.

Lines 220-231: I agree that $\delta^{15}\text{N}$ values of $\sim 2 \text{‰}$ suggest an enhanced contribution from N fixation, but this is independent of ocean (or atmospheric) redox. Phanerozoic OAE's generally have much lower $\delta^{15}\text{N}$ values (down to -5‰ ; Jenkyns et al., 2007, 2001; Rau et al., 1987; Higgins et al., 2012; Kuypers et al., 2004; etc), interpreted as reflecting: 1) enhanced N fixation due to P release from sediments under low O₂ bottom water conditions; 2) larger fractionations from cyanobacteria fixing N at the high [Fe] typical of low-O₂ waters; and/or 3) ammonium uptake. More typical $\delta^{15}\text{N}$ values of $\sim 2 \text{‰}$ are characteristic of modern aerobic marine settings dominated by N₂ fixation, such as in open ocean settings where nutrient upwelling is minimal, and do not require an anoxic water column or

enhanced ammonium availability (in fact they argue against the latter). Owing to the high-energy requirements, microbial nitrogen fixation occurs only in environments in which biologically available nitrogen is deficient relative to other limiting nutrients, which is common in anoxic marine systems. The newly generated $\delta^{15}\text{N}_{\text{kerogen}}$ values are close to 0 ‰, implying that both ammonium and nitrate/nitrite were deficient. Such low $\delta^{15}\text{N}$ values are typical of anoxic sedimentary basins (see Junium & Arthur, 2007). We do not mean to insinuate that specific location was anoxic. In fact, given the Fe-speciation data in Zerkle et al., the water column over core EBA-2 contained some, albeit low levels of dissolved oxygen. We agree with R1, $\delta^{15}\text{N}_{\text{kerogen}}$ would have been lower than 0 ‰ when ammonium is abundant during photosynthesis. We argue that ammonia would have been abundant at/around the GOE. In upper part of the section the accumulation of O_2 would have ensured efficient removal driving diazotrophy. We concede in the manuscript that some of these ideas need to be tested but similar trends in regionally separated cores speaks to a more complex evolutionary pathway of nitrogen cycling than previously thought.

Line 223: What do you mean by a nitrogen cycle “typical of modern marine anoxic environments”? Anaerobic N cycling in modern ocean sediments and OMZ’s is dominated by denitrification and anammox, the isotopic fingerprint of which is evident throughout the entirety of this section, and is the opposite of what you see here. As above, I agree that $\delta^{15}\text{N}$ values of +2‰ suggest that nitrogen fixation was the dominant process in this part of the section, but that could be due to high levels of primary productivity, high P input via weathering, and/or even a change in depositional environment, none of which are redox dependent. We apologise for the confusion. As we have responded to previous comments, we feel that the initial draft was not written with the care it ought to have been, which resulted in several non-descriptive phrases such as that questioned by R1 above. We have removed, or attempted to be explicitly clear, in the redrafted manuscript. We agree with R1 that the lower $\delta^{15}\text{N}$ values at the top of the examined succession reflect diazotrophy and may represent the redistribution of nutrients in the wake of planetary oxygenation. See R1.1, above.

REVIEWER 2:

We thank reviewer two for his/her comments and suggestions. As before, we deal with R2’s comments (orange) in turn, responding in black.

I spent a lot of time thinking about this paper. The results are curious. One of the strongest geochemical signals that has come to define the GOE is the onset of the mass dependent isotopic fractionation of S. That signal has been interpreted as the oxygen dependent production of stratospheric ozone. The build-up of oxygen in the atmosphere had to have altered the N cycle and the data presented here clearly show a positive isotopic fractionation in the three cores. However, unfortunately, the data don’t show that the signal either rose from a lower value deeper in time, or that it underwent fluctuations—except after the GOE. It is hard to reconcile oxygen “oases” when the atmosphere was showing signs of ozone. We use the term ‘oxygen oases’ to refer to discrete localities in pre-GOE oceans where oxygenic photosynthesis increased the dissolved oxygen content. An anoxic atmosphere and deep ocean, testified by the deposition of S-MIF and BIFs, respectively, the oxygen concentration in the surface ocean would be very low (less than a couple of micromoles), limiting the rate of ammonium oxidation (nitrification) (Fennel et al., 2005). Owing to the large fractionation imparted during nitrification and the preferential conversion of ^{14}N to nitrate, the residual ammonium would have become enriched in ^{15}N , which we infer as the mechanism responsible for the high $\delta^{15}\text{N}$ values during the S-MIF interval (pre-GOE) and perhaps thereafter. There is sound evidence that this process started much earlier, and its earliest occurrence places a minimum age-constraint on the emergence of oxygenic photosynthesis; however, this is not the focus of our study. Instead, our $\delta^{15}\text{N}$ data suggest that the slight increase in atmospheric O_2 at the beginning of the GOE did not substantially alter the nitrogen cycle, although it was large enough to terminate the S-MIF signals. Localised positive $\delta^{15}\text{N}$ values have been reported from at least the Neoproterozoic. Instead, we argue that, it was the continued atmospheric accumulation of O_2 that

prompted the evolution of the nitrogen cycle—from partial nitrification (before the GOE or during the early GOE) to complete nitrification (late GOE). As the product (nitrate/nitrite) was reduced to N_2 , through a combination of denitrification and anammox, microbial nitrogen fixation became the main source of biologically available nitrogen, inducing the shift toward the lower $\delta^{15}N$ values we observe in the upper Timeball Hill Formation.

The authors clearly understand the fundamental issue, and spend a significant amount of effort to (1) justify why they didn't measure the N isotopes specifically in a kerogen fraction (i.e., that the results are not an artifact), and (2) account for the lower isotopic fractionation after the GOE. Obviously, it is very hard to justify the use of bulk N isotopes without even calibrating the data against a couple measurements made on kerogens—but I will assume for argument's sake that the measurements are as valid as can be expected. That said, what is not clear is how the rise of N could not have led to denitrification and a reduction in carbon burial. The authors should read (or reread) the Fennel et al. paper (<https://marine.rutgers.edu/~kfennel/Fennel%20et%20al%202005%20AJS.pdf>). If the model presented by Fennel et al., is wrong, what happened that led to the GOE? If the model is correct, then why do we start out at such high ^{15}N values in the three cores? Following the concern of multiple reviewers, we generated new kerogen $\delta^{15}N$ data ($\delta^{15}N_{\text{kerogen}}$) for all three study cores to the revised manuscript. See R1.1 for a full expansion.

We agree with R2 that the data are indeed intriguing and the new $\delta^{15}N_{\text{kerogen}}$ data are consistent with the first two stages of the C-O-N model of Fennel et al. (2005). The first stage corresponds to the upper Rooihogte and lower Timeball Hill formations, when the atmosphere was either anoxic ($<10^{-5}$ PAL; pre-GOE) or weakly oxygenated (immediate aftermath of the GOE). Here, ammonium oxidation occurred slower than denitrification, and the generated nitrate was quantitatively denitrified to N_2 . For this reason, ^{15}N -enriched ammonium was the dominant form of biologically available nitrogen, causing the positive $\delta^{15}N_{\text{kerogen}}$ preserved in this interval. The second stage corresponds to the upper Timeball Hill Formation, in which neither nitrate nor ammonium was present in stable. This stage represents a further increase in atmospheric pO_2 , leading to quantitative oxidation of ammonium, although the persistence of widespread oceanic anoxia/suboxia caused all the nitrate to be reduced to N_2 through denitrification and anaerobic ammonium oxidation (anammox). Thus, microbial nitrogen fixation was the dominant source of biologically available nitrogen. The resulting nutrient nitrogen famine must have been overcome by enhanced nitrogen fixation. This is the working hypothesis presented in our newly drafted manuscript. The last stage inferred by Fennel et al. (2005) might have begun in the uppermost Timeball Hill Formation, which requires further investigation.

REVIEWER 3:

Reviewer three provided an overview with more general responses. As before we deal with them in order with their text in orange and our response in black.

Overview and Major Concerns:

This study mostly focuses on deciphering the evolution of the nitrogen biogeochemical cycling across the GOE using nitrogen isotopes analyses of bulk rocks and organic matter carbon isotopes. These results are compared to previous interpretations placing the oxygenation of Earth atmosphere following the disappearance of MIF of sulfur isotopes within the studied sections. Using isotopic signatures the authors developed a multi-step model of evolution of the nitrogen biogeochemical cycle from ammonium oxidation in oxygen oases to complete denitrification and finally modern style cycle 70 Ma after the GOE. The carbon isotopes variations are interpreted to underline aerobic methanotrophic metabolism coupled to ammonium oxidation as previously proposed in the literature for older rocks deposit (ca. 2.7 Ga). The novelties of this study lie on the first record of nitrogen isotopes across a well-studied sedimentary deposit encompassing the GOE (based on MIFS). The paper is well written, illustrations however could/might be improved. Comparison with

previous studies and bibliography are well done and underline the high potential of nitrogen isotopes to bring further insights on the “oxygenation” and life history on Earth.

We thank Dr. Thomazo for his encouragement. Although we now acknowledge that this study is not the first to report coupled N- and C-isotopes across the GOE (see Zerkle et al., 2017), this is the first comprehensive study that combines both isotope systems with quadruple S-isotope systematics, and hence track planetary oxygenation. Importantly, this is the first study that reports N-isotopes in the wake of the GOE and therefore provides the first insight into the N-cycle’s response to planetary oxygenation. We provide an alternative interpretation for the N-isotope record, and suggest that the N-cycle featured an intermediate step and the inception of a truly modern N-cycle lagged planetary oxygenation by c. 70 million years.

The reading of this manuscript raises one major question that, in my opinion, need to be addressed and more deeply discussed. It concerns the C/N ratio reported in Fig.3 and data tables. I do agree that thermal metamorphism is expressed in the reported rock record and that high C/N ratio reflects thermal devolatilisation of N with little isotope change (lines 128–132). However, the low (below Redfield!) C/N needs to be explained a little bit more. If it does indeed reflect adsorption on phyllosilicate it should be reflected in the lithology and a relationship between potassium content and C/N ratios ($\delta^{15}\text{N}$ to testify as well) for instance is expected. If some samples show stronger nitrogen removal than carbon, and the other way around, they should be discussed and tested for secondary alteration independently. Since speciation is also different the reader is expecting a more robust demonstration that secondary processes do not impact the record (4 to 5 per mille shift is in the range of expected secondary effects of combined devolatilisation in one hand and remineralization on clay on the other hand). At least I suggest sorting the data of Fig.3B with respect to lithology and possible nitrogen speciation to discuss secondary processes. One example of this form table data:

KEA-4 551.04-551.13-B 551,08 8,8 1388,2 -32,6 0,43 3,6

KEA-4 551.04-551.13-A 551,08 7,5 645,5 -35,0 6,94 125,4

If I understand, it is slabs of the same sample where the C/N difference is very high (and so speciation probably), the $\delta^{15}\text{N}$ also quite variable 1.3 per mille (note by the way that point is needed not coma). How do you explain this? Did you analyze a piece of mica in the sample A? This needs to be clarified and placed in a context of what the data reflect in terms of primary and secondary processes.

The bulk nitrogen mainly consists of two endmembers, kerogen-nitrogen and ammonium in association with clay minerals. Many factors might have affected the content of the nitrogen trapped in clay minerals, such as the mineralogy of the clay mineral assemblage, the sources of the clay minerals, and any secondary sources of ammonium. As noted by the other two reviewers, the nitrogen isotopic composition of kerogen is a robust recorder of oceanic nitrogen cycling in deep time. In the revised manuscript, we have generated new data on nitrogen isotopic compositions of kerogen ($\delta^{15}\text{N}_{\text{kerogen}}$) isolated from most of the samples. The $\delta^{15}\text{N}_{\text{kerogen}}$ data show similar trends to $\delta^{15}\text{N}_{\text{bulk}}$, although a small offset is present between $\delta^{15}\text{N}_{\text{kerogen}}$ and $\delta^{15}\text{N}_{\text{bulk}}$ in some of the samples. These new $\delta^{15}\text{N}_{\text{kerogen}}$ data support our interpretations and suggest that the bulk data have not been significantly overprinted. See R1.1 for a more detailed expansion.

This said, I agree with the conclusions even if some remain rather theoretical. I highly recommend to depict suggest changes in the biogeochemical cycling of nitrogen through time by adding a step cartoon representing changes in speciation and reactions. Thank you for the suggestion. We have added a cartoon figure (Figure 5 in the revised manuscript) showing the characteristics of nitrogen cycling during different time intervals.

This is overall a good and interesting contribution for Nat.com journal, which will find easily a broad audience after some clarification. Again, thank you for your encouragement.

Specific comments:

Line 24: high-resolution is overtake. We have avoided subjective terms like this throughout the new manuscript.

Line 38: It's not clear what "substantive oxidative nitrogen cycle" means. You and others described oxidation reactions well before the GOE (cf. line 159) and the reaction you suggest after the GOE is characterizing the cycle in "modern anoxic environment". A cycle where nitrification could be quantitative may be but I suggest to stay on that bottom line to avoid misunderstanding. We have altered the text throughout the manuscript to remove the non-descriptive term "modern aerobic N-cycle" and its derivatives, or explicitly outlined what we mean. Others have inferred that modern nitrogen cycling operated once positive N-isotope ratios were recorded in the rock record. In fact, these positive values only mean that reactions requiring oxygen were ongoing—whether that be partial nitrification or reductive loss via denitrification. Both happen in a modern N cycle; however, the latter dominates. We argue that the former was more important at/around the GOE with a transition to the latter following a significant time lag.

Line 69: may cite here Ader et al. (2016). It is a review paper on that question. Citation added.

Line 76: Again, you may refer to Ader et al. (2016) that compile $\delta^{15}\text{N}_{\text{bulk-Kero}}$ offsets for Precambrian sedimentary rocks. Citation added.

Line 165: At least a ref for this inference. This inference is deleted in the new manuscript.

Line 193: Add the precise age of the Tumbiana Fm. The precise age was added.

Line 210: argues against incomplete turn to - argue for complete. Incomplete denitrification was argued for in other papers (e.g., Garvin et al., 2009). Our data do not support this interpretation as denitrification would be quantitative in weak oxidized environment (dissolved O_2 concentration < 11 μM). We propose that partial ammonium oxidation was at least partially responsible for the positive $\delta^{15}\text{N}_{\text{kerogen}}$ values.

Line 219: Add ref for the "typical" $\delta^{13}\text{C}_{\text{Org}}$ values (and a delta is missing). Des Marais, 1997 was added in the revised manuscript.

Line 209: The question of the mass balance is interesting but ammonia oxidizing bacteria and methanotrophs able to co-oxidize ammonia and methane have different fractionation (around 20 compare to 55 per mille, respectively). We concede that ammonium oxidation can be conducted by methanotrophs; however, methanotrophs create extremely ^{13}C -depleted biomass. We see no evidence for methanotrophy in the $\delta^{13}\text{C}_{\text{Org}}$ records in these cores and hence we seek an alternative explanation. The large differences in the magnitude of the $\delta^{15}\text{N}_{\text{kerogen}}$ data reported here are much smaller than those of the Tumbiana Fm., which may reflect the metabolisms of different microbial consortia.

In your stepped scenario, there is never a biogeochemical cycling allowing for a growing nitrate oceanic reservoir because denitrification is always quantitative—so "modern style nitrogen cycle" is quite an overtake. We apologize, as before, we meant the modern-style nitrogen cycle in an anoxic stratified ocean. To prevent potential misunderstanding, we changed 'modern style nitrogen cycle' to 'nitrogen cycle characterised by quantitative nitrification and denitrification'. All other instances where confusion might ensue have also been clarified.

Line 239: Once again oxidative is not appropriate I think, after all there is no evidence for a substantial ox-nitrogen species reservoir in the whole dataset. We have deleted this in the revised manuscript.

References cited

Bekker, A. et al. Dating the rise of atmospheric oxygen. *Nature* **427**, 117-120 (2004).

- Bekker, A. & Holland, H. D. Oxygen overshoot and recovery during the early Paleoproterozoic. *Earth and Planetary Science Letters* **317–318**, 295-304 (2012).
- Claire, M.W. et al. Modeling the signature of sulfur mass-independent fractionation produced in the Archean atmosphere. *Geochimica et Cosmochimica Acta* **141**, 365-380 (2014).
- Des Marais, D. Isotopic evolution of the biogeochemical carbon cycle during the Proterozoic Eon. *Organic Geochemistry* **27**, 185-193 (1997).
- Fennel, K., Follows, M. & Falkowski, P. G. The co-evolution of the nitrogen, carbon and oxygen cycles in the Proterozoic ocean. *American Journal of Science* **305**, 526-545 (2005).
- Garvin, J., Buick, R., Anbar, A., Arnold, G. & Kaufman, A. Isotopic evidence for an aerobic nitrogen cycle in the latest Archean. *Science* **323**, 1045-1048 (2009).
- Hardisty, D.S. et al. An iodine record of Paleoproterozoic surface ocean oxygenation. *Geology* **42**, 619-622 (2014).
- Izon, G. et al. Biological regulation of atmospheric chemistry en route to planetary oxygenation. *Proceedings of the National Academy of Sciences* **114**, E2571-E2579 (2017).
- Jenkyns, H., Matthews, A., Tsikos, H. & Erel, Y. Nitrate reduction, sulfate reduction, and sedimentary iron isotope evolution during the Cenomanian-Turonian oceanic anoxic event. *Paleoceanography* **22** (2007).
- Junium, C.K. & Arthur, M.A. Nitrogen cycling during the Cretaceous, Cenomanian-Turonian Oceanic Anoxic Event II. *Geochemistry, Geophysics, Geosystems* **8**, 1-18 (2007).
- Kipp, M.A., Stüeken, E.E., Bekker, A. & Buick, R. Selenium isotopes record extensive marine suboxia during the Great Oxidation Event. *Proc. Natl. Acad. Sci. USA* **114**, 875-880 (2017).
- Luo, G. M. et al. Rapid oxygenation of Earth's atmosphere 2.33 billion years ago. *Science Advances* **2**, e1600134 (2016).
- Rasmussen, B., Bekker, A. & Fletcher, I.R. Correlation of Paleoproterozoic glaciations based on U–Pb zircon ages for tuff beds in the Transvaal and Huronian Supergroups. *Earth and Planetary Science Letters* **382**, 173-180 (2013).
- Stüeken, E., Buick, R. & Schauer, A. Nitrogen isotope evidence for alkaline lakes on late Archean continents. *Earth and Planetary Science Letters* **411**, 1-10 (2015).
- Thomazo, C., Ader, M. & Philippot, P. Extreme ¹⁵N-enrichments in 2.72-Gyr-old sediments: evidence for a turning point in the nitrogen cycle. *Geobiology* **9**, 107-120 (2011).
- Waldbauer, J.R., Newman, D.K. & Summons, R.E. Microaerobic steroid biosynthesis and the molecular fossil record of Archaean life. *Proc. Natl. Acad. Sci. USA* **108**, 13409-13414 (2011).
- Zerle, A. L. et al. Onset of the aerobic nitrogen cycle during the Great Oxidation Event. *Nature* **542**, 465-467 (2017).

Reviewers' comments:

Reviewer #1 (Remarks to the Author):

The authors have substantially revised the manuscript, and in the process greatly improved on the description of the cores, discussion of previous work, and exploration of data fidelity. However, there are still some inconsistencies throughout the discussion and interpretation of the data that need to be addressed prior to publication. In addition, as currently written the manuscript primarily focuses on a set of strawman hypotheses, based on misrepresentation of previous work and models that are somewhat out-of-date. For example, the authors present alternative hypotheses for previous interpretations of data from part of this same section – they have some interesting and valid points, but should be more consistently balanced in their consideration of the possible interpretations. As written, they are pushing their alternative hypotheses unconvincingly, without any real justification or additional supporting data. The authors would do better to refocus the manuscript to emphasize what is new and exciting about their own data, namely the extended nature of their records and the changes recorded in the upper part of the section, which could be highly relevant to proposals of nutrient limitation (or not) throughout the mid-Proterozoic.

Detailed comments follow:

Lines 25, 29: the abstract states that aerobic N cycling was established prior to the GOE, but then that the modern N cycle *sensu stricto* lagged the GOE by 70 million years – these statements seem contradictory. What do you mean by “*sensu stricto*” in this case? The authors seem to be defining this as a nitrogen cycle including aerobic processes operating at atmospheric and/or marine oxygen levels similar to the modern ocean. I am not aware that any previous study has claimed that this was occurring anytime in the Precambrian, certainly not throughout the Proterozoic, and indeed the complex spatial variability of the N cycle even in modern environments make it difficult to define a modern N cycle “*sensu stricto*”. This is one of the strawman hypotheses that is pervasive throughout the manuscript.

Lines 60-64: Here and throughout the manuscript, the Garvin and Godfrey papers are not an appropriate reference for the minimum age of oxygenic photosynthesis, as there are MANY other lines of evidence used to constrain this (e.g., see review in Farquhar et al., 2011), and the d15N record is not straightforward. These papers are more appropriate for discussing early evidence of transient aerobic N cycling.

Lines 70-81: Fennel et al. (2005) is a modelling paper, and is not based on any data. There are numerous issues with the simplified model they used (it greatly simplifies the N cycle, includes no internal recycling of N but only sources and sinks, assumes a fully sulfidic Canfield ocean, etc.), and a number of more recent manuscripts have proposed similar conceptual models for stepwise evolution of the marine N cycle (e.g., Canfield et al., 2010, *Science*). This is not even the most recent model of feedbacks between N and O in the Proterozoic (e.g., see Olson et al., 2016, *Frontiers in Microbiology*). The most enduring concept Fennel et al. first proposed was the link between N cycling and O₂ in the mid-Proterozoic (and I think they are more correct in their feedback mechanisms than Olson et al). The implications of the current study for that hypothesis are mentioned (and are quite interesting), but could be much more thoroughly explored.

Line 95: “Stability” would imply a chemical equilibrium process, but nitrification/denitrification are biological processes – this does not make sense. Please clarify.

Line 97: These values are only communicated to the geologic record once the DIN species are taken up by biology and buried as biomass.

Lines 105-107: I agree that this work compliments the previous study, and this is the tone that the authors should take throughout the manuscript, rather than setting this up as a strawman argument (as they do later in the MS).

Lines 122-123: I would argue that the oxygen overshoot is far from well-established.

Lines 181-182: also sorbed, not just entombed.

Lines 184-189: discussion of potential for detrital or diagenetic alteration addition of ammonium – this could be easily tested by plotting d15N bulk data versus %K, as I suggested in the previous review.

Lines 193-195: statement that kerogen d15N are more likely to preserve primary marine signals – this is not universally accepted (e.g., see discussion in Stueken et al., 2017, GCA).

Line 203: if you use the Zhang reference, the fractionation during N₂ fixation with alternative nitrogenase enzymes goes down to -7‰, although these are unlikely to have been present in early marine environments.

Line 206: N₂ fixing biomass can go > 0permil, see Tables 1 and 2 in the reference you cite.

Lines 207-208: there are small diagenetic fractionations of up to 4‰ associated with remineralization (e.g., Freudenthal et al., 2001, GCA), particularly under anoxic conditions with partial nitrification leading to enrichments in the residual d15N.

Line 211: cite Hoch et al., 1992, L&O for fractionations during NH₄-uptake.

Lines 215-219: given the above comments, be more specific here, i.e., positive d15N values > 3 or so permil in well preserved sedimentary rocks that have not undergone significant metamorphism.

Line 224: This statement does not make sense. Either aerobic N cycling was happening (probably temporally and spatially constrained to redox interfaces, as previous work has suggested) or it wasn't. What is a "partially" aerobic N cycle?

Lines 231-233: This is not the only interpretation of these small positive values (which were measured in deepwater sediments!!) - see Busigny et al., 2013, Chemical Geology; Zerkle et al., 2017, Nature.

Lines 236-237: "aerobic nitrogen cycling, and thus dissolved nitrate availability, must have been confined to sites of oxygenic photosynthesis" – I agree with this statement, but it contradicts the statement about transient aerobic N cycling above, as this is presumably one reason these small increases in d15N preserved in DEEP WATER SEDIMENTS cannot be directly tied to nitrate cycling.

Lines 257-258: Again, define what you mean here by "sensu stricto". If it is that the O₂ concentration of the atmosphere was different, that seems like a very obvious strawman/hook. Even the modern N cycle is redox-stratified, with much of the action happening in OMZ's, same

as what has been proposed previously. Also, Zerkle et al. did not argue for a strictly modern N cycle like what you are suggesting, this is somewhat misleading.

Lines 262-266: This would be an appropriate point to cite Fennel et al., 2005.

Lines 273-274: I agree, but we don't see this in the data. We see expression of this isotope effect. Therefore, there is no (data-based) evidence to support this model scenario. Please base your interpretations on data (which should inform the models), not the other way around. This whole paragraph/interpretation section is highly speculative, and has no grounding in your (or anyone else's) data.

Lines 298-300: Explain this statement about "teleconnection" - the isotopic systems examined here are all responding to biology, not directly to the atmosphere.

Lines 312-314: This statement about previous Fe speciation data is misleading – these data were predominantly anoxic in the lowermost 40 m of the core, but the remainder (upper 140 m) indicated dominantly oxic deposition.

Line 334: the discussion of the relevance to Fennel et al., could be greatly expanded here, as suggested above.

Lines 347: Also note the N famine relies on diazotrophy not being able to keep up with N loss (generally this is attributed to metal limitation; but see Zerkle et al., 2006, *Geobiology*) - perhaps your data suggest that N fixation was able to keep up fine?

Lines 350-354: This is the most interesting point made by the manuscript – I would like to see this more thoroughly discussed/explored.

Line 371: Again, Zerkle et al. never argued that this change was unidirectional, only that their data constituted the first evidence for widespread nitrate availability, the isotopic imprint of which remained based on currently available data.

Lines 373-375: Regardless of what model you "prefer", interpretations should be based on data. It is really unclear from this manuscript what data point to a modern-type N cycle 70 million years later. This statement is made in both the abstract and conclusions, but needs to be much more clearly justified in the discussion and interpretation of the data.

Figure 5: This is a lovely figure, but I'm not convinced the data support it. Also note that widespread deep ocean oxygenation did not occur until the Neoproterozoic (e.g., see Canfield et al., 2007, *Science*, and many subsequent papers).

Reviewer #2 (Remarks to the Author):

I have examined the revised paper and the letter and am satisfied with the authors' responses to my concerns.

Reviewer #3 (Remarks to the Author):

Review of the paper entitled: Dynamic Evolution of the Marine Nitrogen Cycle in Response to Planetary Oxygenation

Written by Genming Luo et al.

I really appreciate the reading of the revisions of the Luo et al. manuscript.

The manuscript has been deeply revised and a considerable amount of work has been done in all aspects including technical (i.e. new database including kerogen), writing, argumentation and illustration.

The hypothesis argued in the manuscript is timely and developed with a great care of referring to previous work and hypothesis. It is based on well study cores with multiple sedimentological, dating, geochemical studies already performed on them that fit well with the hypothesis developed by the authors. Although their working hypothesis is different from Zerkle et al., 2017 study, it do not think that it precludes the publication of this interesting (and extended compared to Zerkle et al., 2017) database. The interpretations are well argued, well written and congruent.

There is two points I would liked the authors addressed before publications and some minor suggestions listed below.

First, lines 251-254 the authors acknowledge that most studies reporting positive $\delta^{15}\text{N}$ during the Archean suggest complete nitrification and incomplete denitrification. However, nowhere it is stated clearly that some also already suggest the partial nitrification mechanism as a possible driver for high Archean $\delta^{15}\text{N}$. This is however clearly suggested in (at least) one paper I know well! (i.e. Thomazo et al., 2011).

"This implies that the oxidant availability increased up to the point of allowing ammonia oxidation to N_2O by the co-oxidation of ammonia and methane by methanotrophs (Mandernack et al., 2009) or to nitrite by Ammonia Oxidizing Bacteria (AOB) and Archaea (AOA) but that the oxidant availability was limited relative to ammonia. In these conditions, both nitrite and nitrate would have been immediately and quantitatively reduced to N_2O and/or N_2 by denitrification s.l., leaving ammonium as the main inorganic fixed nitrogen species in the system."

It would be nice to give a better credit to this previous work and to acknowledge that the idea of incomplete nitrification (whatever it is with co-oxidation of methane or not) followed by complete denitrification is a mechanism that has also been argued for a much older rock record! The previously published inference might be wrong (Stueken et al., 2015) but a few words of what would mean having these mechanisms earlier on would be nice too.

The second important point concerns the database. I do really appreciate the efforts to produce kerogen nitrogen results and I have some questions raised by the new table. First, I did not find any caption so may be I misunderstand something. Indeed, when I tried to compare N concentrations of bulk and kerogen phases I end up with some trouble because there is a non-negligible amount of samples with concentrations of N bigger at bulk level than in the residue as N_{ker} . It is surprising to my opinion and it would be great to add the wt.% of the HF residue and to try to balance N concentrations (and also eventually to performed an isotopic mass balance

though). All N concentrations might be in ppm as well for more direct comparison.

Minor points:

- line 99 precise the age and on which samples

- line 160 needs a ref (e.g. Johnston et al., 2012 for Proterozoic examples?

Doi: 10.1038/nature10854 but they argued using the D13C only the d13Corg is may be to less for this inference)

- line 165 introduce range and average per cores

- line 171 reference to the pioneering work from Gray Bebout would be nice here

- lines 251-254 see above and temperate by adding that other mechanism exist in the literature

- line 259 composition of the

- line 403 10 wt.% or 10% of the measured value?

- line 409 would be nice to know how many liter of HF solution was needed and what left over residue after this step

Fig. 3 some small symbol sizes are very small! Please homogenize

Fig. 5 cartoon C – with given arrow the result in terms of d15N is the same than panel B is it what you mean? I guess nitrate uptake is missing, also I think you could remove the nitrate reservoir that is in the anoxic part and simply used the nitrate form above that is partially denitrified at the interface.

Table 1 All N concentrations in the same unit (ppm may be) and caption?

It was a pleasure to become more acquainted with your research

C. Thomazo

RESPONSES TO THE REVIEWERS COMMENTS:

As detailed in the cover letter, buoyed by the positive nature of the reviewers' comments, we revised our manuscript for consideration for publication in *Nature Communications*. Following our first substantial revision we received three sets of reviewers' comments. One of the reviewers was completely satisfied by our resubmission, while the remaining two expressed varying degrees of reservation. As part of this redraft, we have considered the comments of these two reviewers and implemented changes where necessary. Here we detail our responses to the reviewers' comments on a point-by-point basis, which we organize by the order in which we received them. Their original text is provided first in orange followed by our responses in black. Thank you again for your willingness to consider this resubmitted manuscript. We feel that this redraft is much-improved and is substantially clearer to specialists and non-specialists alike.

NOTE: The line numbers referenced below correspond to the revised manuscript without track-changes.

REVIEWER 1:

The authors have substantially revised the manuscript, and in the process greatly improved on the description of the cores, discussion of previous work, and exploration of data fidelity. However, there are still some inconsistencies throughout the discussion and interpretation of the data that need to be addressed prior to publication. In addition, as currently written the manuscript primarily focuses on a set of strawman hypotheses, based on misrepresentation of previous work and models that are somewhat out-of-date. For example, the authors present alternative hypotheses for previous interpretations of data from part of this same section – they have some interesting and valid points, but should be more consistently balanced in their consideration of the possible interpretations. As written, they are pushing their alternative hypotheses unconvincingly, without any real justification or additional supporting data. The authors would do better to refocus the manuscript to emphasize what is new and exciting about their own data, namely the extended nature of their records and the changes recorded in the upper part of the section, which could be highly relevant to proposals of nutrient limitation (or not) throughout the mid-Proterozoic.

We thank the reviewer 1 (R1) for agreeing to review our manuscript for a second time. Following R1's suggestions, we have tried to improve the balance, arguments and reasoning presented in our manuscript. Although R1 apparently favours an interpretation that differs from the one we proposed, we have made our reasoning explicitly clear. In fact, we suggest that both interpretations are unlikely to be mutually exclusive and both probably contributed, with their balance evolving over space and time. Again, this is made explicitly clear in the resubmitted manuscript (lines 257–294). We are grateful to R1 for pointing out our oversights, which were corrected and detailed below.

The issues R1 raises relate predominantly to our interpretation of positive $\delta^{15}\text{N}$ values seen in the upper Rooihogte and lower Timeball Hill formations. This differs from Zerkle *et al.* (2017). We hypothesize that the positive $\delta^{15}\text{N}$ values were the product of partial ammonium oxidation (nitrification), whereas Zerkle *et al.* (2017) argue they result from partial denitrification. We feel the statement that we “*are pushing their alternative hypotheses unconvincingly, without any real justification or additional supporting data*” was unjustified. As before, we previously stated that both were likely possible stating: “*In reality, of course, both mechanisms are not mutually exclusive, and their relative dominance probably varied spatially and temporarily.*” Nevertheless, we argue that given the low oxygen backdrop (supported by S-MIF, I/Ca, $\delta^{82/78}\text{Se}$ etc.) the nitrification rate may have been

much lower than the denitrification rate (Fennel et al., 2005). Thus, the isotopic effect of nitrification could well have been expressed. This is different to the operation of the present-day nitrogen cycle and is worthy of discussion. Throughout the revised manuscript we have rewritten portions of the text to ensure that our discussions are balanced and clear.

In order to address R1's concern about the current manuscript "primarily focuses on a set of 'strawman hypotheses', based on misrepresentation of previous work and models that are somewhat out-of-date", we have corrected and moderated our language when referring to previous work throughout the text. We stress that our study is the most comprehensive nitrogen isotope study conducted in temporal proximity to the GOE to date, and refocused the manuscript to emphasize what is new and exciting about our data: the extended nature of our records and the changes recorded in the upper part of the section (line 332 onwards). We have now added text to further discuss the novelties of our dataset and the potential implications for the mid-Proterozoic nutrient limitation as suggested (line 347 onwards).

DETAILED COMMENTS:

Lines 25, 29: The abstract states that aerobic N cycling was established prior to the GOE, but then that the modern N cycle *sensu stricto* lagged the GOE by 70 million years – these statements seem contradictory. What do you mean by "sensu stricto" in this case? The authors seem to be defining this as a nitrogen cycle including aerobic processes operating at atmospheric and/or marine oxygen levels similar to the modern ocean. I am not aware that any previous study has claimed that this was occurring anytime in the Precambrian, certainly not throughout the Proterozoic, and indeed the complex spatial variability of the N cycle even in modern environments make it difficult to define a modern N cycle "sensu stricto". This is one of the strawman hypotheses that is pervasive throughout the manuscript.

We apologize for this confusion. We now defined and clarified what we mean in a footnote that has been added to the manuscript. In our original submission, we used "cycling" and "cycle" with different connotations. For example, aerobic nitrogen cycling (the process of oxidizing ammonium to nitrogen oxyanions) contributes to the overall nitrogen cycle (the balance and distribution of these biological processes). Therefore, in this framework, the processes that constitute aerobic nitrogen cycling can exist in a nitrogen cycle that is not modern in its narrowest sense (*sensu stricto*).

As for R1's assumption that we define the "nitrogen cycle including aerobic processes operating at atmospheric and/or marine oxygen levels similar to the modern ocean", is incorrect. We agree with R1 that this confusion plagues the literature on the early Earth's nitrogen cycle. Some authors imply that the 'modern nitrogen cycle' represents the appearance of related metabolic processes (e.g., nitrification, denitrification), with less emphasis on the internal relationships between these metabolisms. We wish to advance beyond this following R1's recommendation. In the revised manuscript, we discussed nitrogen cycle in terms of the likely processes that were active and their relative importance.

To improve the manuscript, we have defined explicitly what we mean and thoroughly revised our terminology accordingly. We have amended the abstract as suggested.

Lines 60-64: Here and throughout the manuscript, the Garvin and Godfrey papers are not an appropriate reference for the minimum age of oxygenic photosynthesis, as there are MANY other lines of evidence used to constrain this (e.g., see review in Farquhar et al., 2011), and the $\delta^{15}\text{N}$ record is not straightforward. These papers are more appropriate for discussing early evidence of transient aerobic

N cycling.

We agree that there are many lines of evidence to constrain the minimum age of photosynthesis, many of which are reviewed in Farquhar et al. (2011), which we cited further down the manuscript (in a more appropriate place). We would like to make clear that the point of this paragraph was to introduce the link between oxygen and the processes that fractionate nitrogen isotopes, which is fundamental for the manuscript, rather than constrain the evolution of oxygenic photosynthesis. We changed the text to ‘Given this intrinsic link between environmental oxygen availability and nitrogen cycling, the timing of the transition from an anaerobic to an aerobic (i.e., featuring metabolisms involving nitrogen oxyanions) nitrogen cycle has been used as a line of evidence to constrain the minimum age of the emergence of oxygenic photosynthesis (Garvin et al., 2009; Godfrey and Falkowski, 2009).’.

Lines 70-81: Fennel et al. (2005) is a modelling paper, and is not based on any data. There are numerous issues with the simplified model they used (it greatly simplifies the N cycle, includes no internal recycling of N but only sources and sinks, assumes a fully sulfidic Canfield ocean, etc.), and several more recent manuscripts have proposed similar conceptual models for stepwise evolution of the marine N cycle (e.g., Canfield et al., 2010, Science). This is not even the most recent model of feedbacks between N and O in the Proterozoic (e.g., see Olson et al., 2016, Frontiers in Microbiology). The most enduring concept Fennel et al. first proposed was the link between N cycling and O₂ in the mid-Proterozoic (and I think they are more correct in their feedback mechanisms than Olson et al.). The implications of the current study for that hypothesis are mentioned (and are quite interesting), but could be much more thoroughly explored.

Reviewer 1 is correct when they state Fennel et al. (2005) is a modelling paper and it is not based on any data. We agree that the model presented by Fennel et al. (2005) is simplified: it’s a parameterized approximation of the nitrogen cycle used to explore and develop a hypothesis. However, many of the boundary conditions of the model required to develop a more complex model are poorly known at present. Therefore, we conduct studies—like this one— to obtain data to refine these models.

The more recent work mentioned by R1 has picked up on the ideas and inferences made by Fennel et al. (2005) but has not significantly advanced it (one is a review and the other a perspective piece). We have added these references to the text as both are supportive of the ideas conceived by Fennel et al. (2005). Moreover, we agree that the implications of these theoretical and computational approaches are interesting and have ramifications for the evolution of the nitrogen cycle and oxygen. That’s why we included it in the first place. Rather than discuss the implications here in the introduction, which is premature given the article’s structure, we have expanded on these specific points in our discussion as suggested.

Line 95: “Stability” would imply a chemical equilibrium process, but nitrification/denitrification are biological processes – this does not make sense. Please clarify.

To avoid confusion, this text has been removed as it repeats information that features in the discussion.

Line 97: These values are only communicated to the geologic record once the DIN species are taken up by biology and buried as biomass.

We have changed the text as suggested. It now reads “Typically, biologically mediated redox cycling exerts the dominant control on the $\delta^{15}\text{N}$ of dissolved nitrogen-species, which is ultimately communicated to the geological record after biological assimilation (Stüeken et al., 2016)”.

Lines 105-107: I agree that this work compliments the previous study, and this is the tone that the authors should take throughout the manuscript, rather than setting this up as a strawman argument (as they do later in the MS).

We think this work not only complements the previous study, but also goes some way to discuss the evolution of the nitrogen cycle and its implications, which is unique and represents an advance in the field. We expanded on the evolution of the nitrogen cycle and its implications and avoided any 'strawman' or speculative arguments throughout the revised manuscript.

Lines 122-123: I would argue that the oxygen overshoot is far from well-established.

Admittedly there is not a wealth of direct geochemical data to support an increase in atmospheric oxygen during the Lomagundi, however, the opposite is also true. In order to address R1's concern, here, and elsewhere in the manuscript, we have not used definitive language. For this example, we wrote "...prelude of the *inferred* oxygen overshoot to have occurred during the Lomagundi Event...". The use of '*inferred*' is the key point. It conveys a degree of uncertainty and indicates this is a finding from the cited literature rather than our own conjecture. We also added additional references to studies that provide evidence of an apparent transient rise in pO_2 .

Lines 181-182: also sorbed, not just entombed.

The text has been amended accordingly. It now reads: "Bulk sedimentary nitrogen consists not only of organic nitrogen but also of ammonium adsorbed to, and bonded within, the interlayers of clay minerals after substitution for potassium."

Lines 184-189: Discussion of potential for detrital or diagenetic alteration addition of ammonium—this could be easily tested by plotting $\delta^{15}N$ bulk data versus %K, as I suggested in the previous review.

This suggestion was made previously. When we received the previous review, we opted to make additional kerogen derived $\delta^{15}N$ measurements to expand our records rather than determine [K] for multiple reasons: (i) We do not have the analytical capacity to make these measurements, (ii) we do not have the financial means to make these measurements externally, (iii) we have limited sample material remaining preventing conventional XRF analysis, (iv) Zerkle et al. (2017) examined the suggested relationship in samples from the same core and found *no* significant relationship between $\delta^{15}N$ bulk data versus [K] and (v) the lack of observational evidence for metasomatism which has been demonstrated to cause significant ammonium addition elsewhere (e.g. Kump et al., 2011). We have, however, added a sentence that speaks specifically to the lack of relationship in Zerkle et al. (2017). Importantly, the close approximation between bulk and kerogen derived $\delta^{15}N$ demonstrates that neither nitrogen-pool has experienced significant post-depositional alteration. This point is made in the text around lines 184-192. For these reasons, we have not added K abundance data.

Lines 193-195: Statement that kerogen $\delta^{15}N$ are more likely to preserve primary marine signals – this is not universally accepted (e.g., see discussion in Stueken et al., 2017, GCA).

When the manuscript was initially submitted Stueken et al. (2017) was not published. This recent paper has gone some-way to resolve whether bulk or kerogen are a better medium from which to make palaeoenvironmental reconstructions. We include the lessons learnt from this paper in the section entitled "*the fidelity of the isotope records*". We have removed the sentence where we favour the interpretation of a specific nitrogen pool over another. This is further justified given the close approximation between $\delta^{15}N_{\text{Bulk}}$ and $\delta^{15}N_{\text{Kerogen}}$.

Line 203: If you use the Zhang reference, the fractionation during N_2 fixation with alternative

nitrogenase enzymes goes down to -7‰, although these are unlikely to have been present in early marine environments.

Thank you for pointing out this oversight. We have kept the Zhang et al. (2014) as it is primary literature. In doing so we have amended the quoted range.

Line 206: N₂ fixing biomass can go > 0 per mil, see Tables 1 and 2 in the reference you cite.

Thanks for pointing this out. Indeed, in detail, when you consider the uncertainty, especially at the 2 sigma-level, there is only one value that convincingly exceeds 1 per mil. This is the Wada (1980) study, cited in Zerkle et al. (2008). We were unable to find this book chapter to verify this data and the appropriate uncertainty was not cited in the compiled table. Given this uncertainty we have slightly changed the text so it is not as explicit and more correct. It now reads: *“Moreover, while microbial culture experiments demonstrate that trace metal availability can alter the magnitude of isotope fractionation associated with diazotrophy, they demonstrate that nitrogen fixation is unlikely producing biomass with a $\delta^{15}\text{N}$ much higher than 0 ‰ (Zerkle et al., 2008; Bauersachs et al., 2009).”*

Lines 207-208: There are small diagenetic fractionations of up to 4 ‰ associated with remineralization (e.g., Freudenthal et al., 2001, GCA), particularly under anoxic conditions with partial nitrification leading to enrichments in the residual $\delta^{15}\text{N}$.

The paper R1 recommends (Freudenthal et al., 2001, GCA) opens its conclusions with *“In our study, the observed $\delta^{13}\text{C}_{\text{Org}}$ and $\delta^{15}\text{N}_{\text{TN}}$ due to early diagenetic processes was less than 1 ‰”,* moreover, rather than focusing on anoxic processes, the paper focuses on deamination in aerobic environments—clearly this is an incorrect reference for the point in question.

This is another point in the manuscript where we use specific language to convey uncertainty—note the word “appreciable”. In fact, the magnitude, not to mention the sign, of the isotope effect depends on the method of remineralisation, the chemical composition of the precursor organic material and the prevailing environmental redox conditions. To convey this uncertainty, we have added “(< 3 ‰)”, citing Stüeken et al. (2016) who review the subject, and show there is no consensus based on the literature they cite.

Line 211: Cite Hoch et al., 1992, L&O for fractionations during NH₄-uptake.

Added, as suggested.

Lines 215-219: Given the above comments, be more specific here, i.e., positive $\delta^{15}\text{N}$ values > 3 or so permil in well preserved sedimentary rocks that have not undergone significant metamorphism.

We revised the text to be more specific here. The text now reads: *“Consequently, the emergence of positive $\delta^{15}\text{N}$ values (> 3 ‰) in well-preserved and unaltered sedimentary rocks from open-marine settings is generally interpreted as the isotopic fingerprint of an aerobic nitrogen cycle...”*.

Line 224: This statement does not make sense. Either aerobic N cycling was happening (probably temporally and spatially constrained to redox interfaces, as previous work has suggested) or it wasn't. What is a "partially" aerobic N cycle?

With hindsight, we agree this is confusing. We amended the text as suggested, and it now reads: *“Therefore, despite the anoxic backdrop, the preferred explanation of these positive $\delta^{15}\text{N}$ values commonly invokes the intermittent operation of an aerobic nitrogen cycle (Garvin et al., 2009; Godfrey and Falkowski, 2009; Thomazo et al., 2011; Zerkle et al., 2017).*

Lines 231-233: This is not the only interpretation of these small positive values (which were measured in deep-water sediments!!) - see Busigny et al., 2013, Chemical Geology; Zerkle et al., 2017, Nature.

We have added the alternate explanations and their sources, as suggested. Please note that we discuss this at length further down the manuscript.

Lines 236-237: “aerobic nitrogen cycling, and thus dissolved nitrate availability, must have been confined to sites of oxygenic photosynthesis” – I agree with this statement, but it contradicts the statement about transient aerobic N cycling above, as this is presumably one reason these small increases in $\delta^{15}\text{N}$ preserved in DEEP WATER SEDIMENTS cannot be directly tied to nitrate cycling.

We added the possibility that the positive $\delta^{15}\text{N}$ values might be ascribed to incomplete ammonium oxidation to N_2 by ferric hydroxides. However, this mechanism is unlikely to be responsible for the positive $\delta^{15}\text{N}$ values in the lower part of the succession because (i) the concentration of iron oxides is very low, and (ii) the sequence studied here was deposited in a relatively shallow marine environment (deltaic environment).

Lines 257-258: Again, define what you mean here by “sensu stricto”. If it is that the O_2 concentration of the atmosphere was different, that seems like a very obvious strawman/hook. Even the modern N cycle is redox-stratified, with much of the action happening in OMZ's, same as what has been proposed previously. Also, Zerkle et al. did not argue for a strictly modern N cycle like what you are suggesting, this is somewhat misleading.

Much of the misunderstanding apparently arises from the misinterpretation of this Latin—which simply equates to in its narrowest sense. We used this terminology, as explained above, to describe the balance that resembles that seen presently where nitrogen oxyanions dominate and denitrification was confined to OMZs as R1 mentioned. We have removed and clarified text to better convey what we originally meant. We also removed the reference to Zerkle et al as they do not explicitly speak to a modern N cycle. The text now reads ‘While we acknowledge that most of the processes central to the modern nitrogen cycle (e.g., diazotrophy, nitrification and denitrification) had almost certainly evolved by the late Archaean (e.g., Stüeken et al., 2015a; Zerkle et al., 2017), their relative importance remains unresolved. This, in turn, leaves an open question: How did the nitrogen cycle evolve in the aftermath of the GOE?’

Lines 262-266: This would be an appropriate point to cite Fennel et al., 2005.

This reference has been added as suggested.

Lines 273-274: I agree, but we don't see this in the data. We see expression of this isotope effect. Therefore, there is no (data-based) evidence to support this model scenario. Please base your interpretations on data (which should inform the models), not the other way around. This whole paragraph/interpretation section is highly speculative, and has no grounding in your (or anyone else's) data.

We have thoroughly revisited the language between lines 239 and 331 to ensure that definitive language has been substituted for language that conveys uncertainty. This now clearly outlines both interpretations, their uncertainties and the potential for evolution. As we are careful to develop, the isotope effect associated with complete-nitrification coupled to partial denitrification is inseparable from that associated with partial-nitrification coupled to complete denitrification—they will both produce positive $\delta^{15}\text{N}$ values. The main difference between these two hypotheses is the oxygenation state of the ocean-atmosphere system. As we discussed in the manuscript, multiple lines of evidence, e.g., large $\Delta^{33}\text{S}$ values and low I/Ca ratios, suggest that the oxygen content in the ocean-atmosphere system was very low. In this case, the denitrification would outpace the nitrification, which nulled the isotope effect of denitrification. Therefore, in our opinion, in most parts of the ocean

partial-nitrification coupled with complete denitrification was the main mechanism inducing the positive $\delta^{15}\text{N}$ values. Meanwhile, we concede that in regions of active photosynthetic activity and O_2 production, nitrate may have been stable and utilised by phytoplankton as their preferred nitrogen-substrate.

Lines 298-300: Explain this statement about “teleconnection” - the isotopic systems examined here are all responding to biology, not directly to the atmosphere.

We have clarified the statement about ‘teleconnection’. The text now reads: *“These chemostratigraphic trends are accompanied by a decrease in $\delta^{13}\text{C}_{\text{org}}$ and $\delta^{34}\text{S}_{\text{pyrite}}$, consistent with increased surface-water oxygen production/availability in turn promoting aerobic nitrogen cycling, oxidation of ^{13}C -depleted carbon phases (e.g., methane or dissolved organic carbon), as well as an increase in the marine sulphate inventory (Luo et al., 2016). These coeval isotopic changes argue for a strong yet evolving connection between the biosphere and the chemical composition of the atmosphere and ocean, whereby the chemical evolution of Earth’s surficial environments influenced the emergence and distribution of ecological niches, not to mention the metabolisms that filled them”.*

Lines 312-314: This statement about previous Fe speciation data is misleading – these data were predominantly anoxic in the lowermost 40 m of the core, but the remainder (upper 140 m) indicated dominantly oxic deposition.

We have added the depths to preclude any ambiguity. The discussion proceeds from the bottom of the record upwards.

Line 334: the discussion of the relevance to Fennel et al., could be greatly expanded here, as suggested above.

As suggested, we have expanded the discussion at the end of the manuscript. See lines 350 onwards.

Lines 347: Also, note the N famine relies on diazotrophy not being able to keep up with N loss (generally this is attributed to metal limitation; but see Zerkle et al., 2006, Geobiology) – perhaps your data suggest that N fixation was able to keep up fine?

As suggested, we have now expanded the discussion at the end of the manuscript. We draw attention to the arguments surrounding trace metal limitation and that our data conflict with those arguments. See lines 350 onwards.

Lines 350-354: This is the most interesting point made by the manuscript – I would like to see this more thoroughly discussed/explored.

We have expanded on our discussion as detailed above; however, it is important to note that this is a hypothesis. Work is on-going to test these ideas from more spatially separated sedimentary successions. See the closing paragraph.

Line 371: Again, Zerkle et al. never argued that this change was unidirectional, only that their data constituted the first evidence for widespread nitrate availability, the isotopic imprint of which remained based on currently available data.

We have removed the clause from between the parenthetical commas. On rereading Zerkle et al. (2017) they don’t explicitly make the statement; we concede that this was our inference based on the statement *“indicating that aerobic nitrogen cycling became at least locally widespread enough to impart a long-lived isotopic imprint on the marine $\delta^{15}\text{N}$ during the GOE”*. The questioned text in the revised manuscript now reads: *“High-resolution $\delta^{15}\text{N}$ studies, particularly in the prelude and aftermath of the Lomagundi event, are required to test our inferences and more completely explore whether the*

evolution of the marine nitrogen cycle was unidirectional, or whether it was spatio-temporally dynamic, as might be predicted from consideration of other oxygen sensitive proxy records (e.g., Lyons et al., 2014; Kipp et al., 2017) “.

Lines 373-375: Regardless of what model you “prefer”, interpretations should be based on data. It is really unclear from this manuscript what data point to a modern-type N cycle 70 million years later. This statement is made in both the abstract and conclusions, but needs to be much more clearly justified in the discussion and interpretation of the data.

Thank you, we appreciate the opportunity to clarify our points in greater detail. We acknowledge that we didn't develop and discuss the derivation of the age constraint in the abstract and conclusions. Specifically, as explained above, a modern-type N cycle was defined previously by the appearance of quantitative nitrification in oxic water, irrespective of whether denitrification was quantitative (e.g., modern stratified basins) or non-quantitative (e.g., modern OMZs). We suggested that the $\delta^{15}\text{N}$ values near 0 per mil in the lower upper Timeball Hill Formation indicate quantitative nitrification and denitrification—Resembling present-day nitrogen cycling in stratified basins. Given the age of the GOE (~2330 Ma) and the U-Pb age of the lowermost part of the upper Timeball Hill Formation (~2260 Ma), we estimated the duration of these two events to be approximately 70 million years. Following the R1's suggestion, we changed this part substantially and the specific duration was not included. Please see the abstract and the last paragraph of the main text.

Figure 5: This is a lovely figure, but I'm not convinced the data support it. Also, note that widespread deep ocean oxygenation did not occur until the Neoproterozoic (e.g., see Canfield et al., 2007, Science, and many subsequent papers).

We fully agree that that widespread oxygenation is a much later phenomenon beginning in the Neoproterozoic. Panel C only represents the Lomagundi interval which might have had higher $p\text{O}_2$ as many have argued (see in text citations). We have altered the figure caption to clarify this issue.

REVIEWER 2:

I have examined the revised paper and the letter and am satisfied with the authors' responses to my concerns.

We thank reviewer 2 for agreeing to review our amended manuscript and our responses and we are glad that our revisions could resolve their concerns.

REVIEWER 3:

I appreciated reading of the revisions of the Luo et al. manuscript.

The manuscript has been deeply revised and a considerable amount of work has been done in all aspects including technical (i.e. new database including kerogen), writing, argumentation and illustration.

The hypothesis argued in the manuscript is timely and developed with a great care of referring to previous work and hypothesis. It is based on well-studied cores with multiple sedimentological, dating, geochemical studies already performed on them that fit well with the hypothesis developed by the authors. Although their working hypothesis is different from Zerkle et al., 2017 study, I do not think that it precludes the publication of this interesting (and extended compared to Zerkle et al., 2017) database. The interpretations are well argued, well written and congruent.

There are two points I would like the authors to address before publication and some minor suggestions listed below.

We would like to express our gratitude to Dr Thomazo, his patience and efforts are greatly appreciated. His diligence has undoubtedly helped to clarify the text, illustrations and the supplementary data table. Where possible, we have made all his suggested changes, which we detail beneath his specific comments below.

MAJOR POINTS:

First, lines 251-254 the authors acknowledge that most studies reporting positive $\delta^{15}\text{N}$ during the Archean suggest complete nitrification and incomplete denitrification. However, nowhere it is stated clearly that some have already suggested that partial nitrification as a possible causative mechanism of high Archean $\delta^{15}\text{N}$. This is, however, clearly suggested in (at least) one paper I know well! (i.e. Thomazo et al., 2011).

“This implies that the oxidant availability increased up to the point of allowing ammonia oxidation to N_2O by the co-oxidation of ammonia and methane by methanotrophs (Mandernack et al., 2009) or to nitrite by Ammonia Oxidizing Bacteria (AOB) and Archaea (AOA) but that the oxidant availability was limited relative to ammonia. In these conditions, both nitrite and nitrate would have been immediately and quantitatively reduced to N_2O and / or N_2 by denitrification s.l., leaving ammonium as the main inorganic fixed nitrogen species in the system.”

It would be nice to give a better credit to this previous work and to acknowledge that the idea of incomplete nitrification (whatever it is with co-oxidation of methane or not) followed by complete denitrification is a mechanism that has also been argued for a much older rock record! The previously published inference might be wrong (Stueken et al., 2015) but a few words of what would mean having these mechanisms earlier on would be nice too.

We apologize for this oversight and agree with the sentiments expressed by Dr Thomazo. We included the reference in question in the initial manuscript but unfortunately it got omitted as part of the extensive redraft of the manuscript received *en route* to its present incarnation.

However, instead of adding the sentence in/around the line numbers suggested above, we have added it later in the discussion (Lines 278-280). In our opinion, the suggested placement disrupts the flow of the text and the point of that specific paragraph. Instead we added it after we introduced our alternate interpretation of positive $\delta^{15}\text{N}$ in the studied succession. We feel that the addition of: “*This mechanism, although not universally accepted (Stüeken et al., 2015b), has been proposed previously to explain the extreme positive $\delta^{15}\text{N}$ values recorded in the 2.7 Ga Tumbiana Formation (Thomazo et al., 2011).*” gives the appropriate credit to Thomazo et al. paper while contributing more completely to our discussion. We have also added it elsewhere in the manuscript where appropriate.

The second important point concerns the database. I do appreciate the efforts to produce kerogen nitrogen results and I have some questions raised by the new table. First, I did not find any caption so may have misunderstood something. Indeed, when I tried to compare N concentrations of bulk and kerogen phases I ended up with some trouble because there is a non-negligible quantity of samples with concentrations of N bigger at bulk level than in the residue as N_{ker} . It is surprising to my opinion and it would be great to add the Wt. % of the HF residue and to try to balance N concentrations (and eventually to performed an isotopic mass balance though). All N concentrations might be in ppm as well for more direct comparison.

We apologize for this oversight; the table should have featured an appropriate caption. This has been added to the revised manuscript and each sheet of the excel file that is submitted alongside it. To

avoid misunderstanding, we expressed bulk and kerogen nitrogen abundances in parts-per-million as requested. Almost exclusively kerogen-nitrogen abundances are higher than bulk-nitrogen abundances.

The few instances where this is not the case might from either (i) elevated inorganic N contents in clay-rich, organic lean, samples, (ii) impure residues featuring heavy minerals (e.g., pyrite) which were not removed during the acid treatment (iii) and/or precipitation of fluorosilicate phases during the HF processing. We cannot unequivocally discriminate between these causes. Observational evidence for pyrite in the acid-treated residues suggests that the second explanation could be an important factor.

While, we've added wt. % of the HF residue, given the impurities of the kerogen isolates, a mass balance-derived nitrogen distribution is unlikely to be accurate in these samples.

MINOR POINTS:

Line 99 precise the age and on which samples.

We have made the suggested changes. The text now reads "*These principles were recently applied to reveal how the GOE impacted nitrogen cycling at ~ 2.3 Ga through the upper Rooihogte and lower Timeball Hill Formations (Zerkle et al., 2017)*".

line 160 needs a ref (e.g. Johnston et al., 2012 for Proterozoic examples? Doi:10.1038/nature10854 but they argued using the D13C only the d13Corg is may be to less for this inference). Apologies for the unsupported sentence.

The reference has been added as suggested.

line 165 introduce range and average per cores.

Rather than introduce this for each individual core, which would take a lot of space and detract from the flow of the text, we've added the average \pm standard deviation (1σ) in parenthesis for the whole dataset, as suggested.

line 171 reference to the pioneering work from Gray Bebout would be nice here.

Added as suggested.

lines 251-254 see above and temperate by adding that other mechanism exist in the literature.

As explained above. We've added the suggested references but not at this specific place. Instead, we included a variation of what was suggested, which features on lines 278-280. As before, we apologize for this omission

Line 259 composition of the

Thank you for spotting this. Corrected as suggested.

Line 403 10 wt.% or 10% of the measured value?

Thank you. The text has been changed and it now reads "Replicate analysis of samples and standards demonstrates a relative precision of better than 10 %".

Line 409 would be nice to know how many liters of HF solution was needed and what was the left-over residue after this step.

As requested, we added the volumes of mixed HF:HCl solution that were used to digest the silicate fraction when isolating the kerogens. The mass of the residue after acid treatment have been added into the data set. Unfortunately, in some of the samples, a fluorosilicate, precipitate was formed during the acid treatment process. The text clearly explains that we did not attempt to remove the non-kerogen fractions as they did not affect the quality of nitrogen isotope data (Stüeken et al. 2017).

The caption to Table 1 also includes these details.

Fig. 3 some small symbol sizes are very small! Please homogenize.

The symbols have been resized as requested.

Fig. 5 cartoon C – with given arrow the result in terms of $\delta^{15}\text{N}$ is the same than panel B is it what you mean? I guess nitrate uptake is missing, also I think you could remove the nitrate reservoir that is in the anoxic part and simply used the nitrate form above that is partially denitrified at the interface. Thanks for the suggestion. We have altered the figure to more closely resemble the text.

Table 1 All N concentrations in the same unit (ppm may be) and caption?

Thank you and we apologize for this oversight. A full caption has been added to the table, both in the excel sheet and in the main manuscript. The data have also been sorted by core and reformatted to be more useful. We expressed the abundances of each nitrogen-bearing phase in parts-per million (ppm).

References Cited

- Bauersachs, T., *et al.* Nitrogen isotopic fractionation associated with growth on dinitrogen gas and nitrate by cyanobacteria. *Limnology and Oceanography* **54**, 1403-1411 (2009).
- Busigny, V. *et al.* Nitrogen cycle in the Late Archean ferruginous ocean. *Chemical Geology* **362**, 115-130 (2013).
- Canfield, D. E., Glazer, A. N. and Falkowski, P. G. The evolution and future of Earth's nitrogen cycle. *Science* **330**, 192-196 (2010).
- Farquhar, J., Zerkle, A. & Bekker, A. Geological constraints on the origin of oxygenic photosynthesis. *Photosynthesis Research* **107**, 11-36 (2011).
- Fennel, K., Follows, M. & Falkowski, P. G. The co-evolution of the nitrogen, carbon and oxygen cycles in the Proterozoic ocean. *American Journal of Science* **305**, 526-545 (2005).
- Freudenthal, T., *et al.* Early diagenesis of organic matter from sediments of the eastern subtropical Atlantic: evidence from stable nitrogen and carbon isotopes. *Geochimica et Cosmochimica Acta* **65**, 1795-1808 (2001).
- Garvin, J., Buick, R., Anbar, A., Arnold, G. & Kaufman, A. Isotopic evidence for an aerobic nitrogen cycle in the latest Archean. *Science* **323**, 1045-1048 (2009).
- Godfrey, L. V. & Falkowski, P. G. The cycling and redox state of nitrogen in the Archaean ocean. *Nature Geoscience* **2**, 725-730 (2009).
- Hoch, M. P., Fogel, M.L., & Kirchman, D.L. Isotope fractionation associated with ammonium uptake by a marine bacterium. *Limnology and Oceanography* **37**, 1447-1459 (1992).
- Johnston, D. T., Macdonald, F. A., Gill, B. C., Hoffman, P. F., & Schrag, D. P. Uncovering the Neoproterozoic carbon cycle. *Nature*, **483**, 320-323 (2012).
- Kipp, M.A., Stüeken, E.E., Bekker, A. & Buick, R. Selenium isotopes record extensive marine suboxia during the Great Oxidation Event. *Proc. Natl. Acad. Sci. USA* **114**, 875-880 (2017).
- Kump, L. R. *et al.* Isotopic evidence for massive oxidation of organic matter following the great oxidation event. *Science* **334**, 1694-1696 (2011).

- Luo, G. M. *et al.* Rapid oxygenation of Earth's atmosphere 2.33 billion years ago. *Science Advances* **2**, e1600134 (2016).
- Lyons, T.W., Reinhard, C.T. & Planavsky, N.J. The rise of oxygen in Earth's early ocean and atmosphere. *Nature* **506**, 307-315 (2014).
- Mandernack, L. W., Mills, C. T., Johnson, C. A., Rahn, T. & Kinney, C. The $\delta^{15}\text{N}$ and $\delta^{18}\text{O}$ values of N_2O produced during the co-oxidation of ammonia by methanotrophic bacteria. *Chemical Geology* **267**, 96-107 (2009).
- Olson, S. L., Reinhard C. T. & Lyons T. W. Cyanobacterial Diazotrophy and Earth's delayed Oxygenation. *Frontiers in Microbiology* **7**, 1526 (2016).
- Stüeken, E. E., Buick, R., Guy, B. M. & Koehler, M. C. Isotopic evidence for biological nitrogen fixation by molybdenum-nitrogenase from 3.2 Gyr. *Nature* **520**, 666-669 (2015a).
- Stüeken, E.E., Buick, R. & Schauer, A.J. Nitrogen isotope evidence for alkaline lakes on late Archaean continents. *Earth and Planetary Science Letters* **411**, 1-10 (2015b).
- Stüeken, E. E., Kipp, M. A., Koehler, M. C. & Buick, R. The evolution of Earth's biogeochemical nitrogen cycle. *Earth-Science Reviews* **160**, 220-239 (2016).
- Stüeken, E. E., Zaloumis, J., Meixnerová, J., Buick, R. Differential metamorphic effects in nitrogen isotopes in kerogen extracts and bulk rocks **217**, 80–94 (2017).
- Thomazo, C., Ader, M. & Philippot, P. Extreme ^{15}N -enrichments in 2.72-Gyr-old sediments: evidence for a turning point in the nitrogen cycle. *Geobiology* **9**, 107-120 (2011).
- Wada, E. Nitrogen isotope fractionation and its significance in biogeochemical processes occurring in marine environments, in *Isotope Marine Chemistry*, edited by Goldberg, E.D. et al., 375-398, Uchida Rokakuho, Tokyo, Japan (1980).
- Zerkle, A.L., House, C.H., Cox, R.P. & Canfield, D.E. Melta limitation of cyanobacterial N_2 fixation and implications for the Precambrian nitrogen cycle. *Geobiology* **4**, 285-297 (2006).
- Zerkle, A.L., Junium, C.K., Canfield, D.E. & House, C.H. Production of ^{15}N -depleted biomass during cyanobacterial N_2 -fixation at high Fe concentration. *Journal of Geophysical Research*, 113 (2008).
- Zerkle, A. L. et al. Onset of the aerobic nitrogen cycle during the Great Oxidation Event. *Nature* **542**, 465-467 (2017).
- Zhang, X., Sigman, D. M., Morel, F. M. M. & Kraepiel, A. M. L. Nitrogen isotope fractionation by alternative nitrogenases and past ocean anoxia. *Proc. Natl. Acad. Sci. USA* **111**, 4782-4787 (2014).

REVIEWERS' COMMENTS:

Reviewer #1 (Remarks to the Author):

Excellent revision, greatly improved clarity and focus. I have no additional concerns and am happy to support publication.

Reviewer #3 (Remarks to the Author):

3rd Review of the paper entitled: Dynamic Evolution of the Marine Nitrogen Cycle in Response to Planetary Oxygenation

Written by Genming Luo et al.

The last revision of this study meets, to my opinion, the standard of Nature Communication Journal.

The manuscript has been greatly amended and reads very well. The data, their interpretations and the in-depth associated discussion make it a nice new piece of work in the Geobiology field. I have no doubt that this article will easily reach a broad audience and I'm looking forward of citing it.

Few minor edits suggestions listed below:

Line 38 – move (e.g. NH₄⁺) before forms

Line 101 – remove “typically”

Line 159 – increase to +7 and to +9‰, respectively?

Line 160 – idem

Line 190 – Ader et al., 2006 study is on natural samples, I suggest to avoid “experimental” as it can introduce some confusion.

Line 215 – replace “preserved” by measured

Line 371 – we suggest (that?)

Line 376 – at a lower resolution (and magnitude?)

Final remark: I have some problem with the final part of the final sentence “...when the nitrogen cycle became ecologically modern” - I have the feeling that in the frame of the theory of Evolution (central to ecology) modern can only mean what is observed at the present day or I misunderstand what you mean by modern here. In any case make sure that one could not, somehow, think that the organisms relevant to the N cycle stop evolving sometimes in the Precambrian!

C. Thomazo

1. Reviewer #1 (Remarks to the Author):

Excellent revision, greatly improved clarity and focus. I have no additional concerns and am happy to support publication.

We are glad that our revisions could resolve the concerns raised by reviewer 1 (R1). We thank R1 for their patience and comments throughout the review process. Their thoughtful reviews, and importantly challenges, throughout have pushed us to think more deeply about the operation of the nitrogen cycle during this critical time interval. Their efforts have undoubtedly improved the quality of this manuscript.

2. Reviewer #3 (Remarks to the Author):

3rd Review of the paper entitled: Dynamic Evolution of the Marine Nitrogen Cycle in Response to Planetary Oxygenation written by Genming Luo et al.

The last revision of this study meets, to my opinion, the standard of Nature Communication Journal. The manuscript has been greatly amended and reads very well. The data, their interpretations and the in-depth associated discussion make it a nice new piece of work in the Geobiology field. I have no doubt that this article will easily reach a broad audience and I'm looking forward of citing it.

We, again, would like to extend our gratitude to Dr Thomazo, his patience and efforts are greatly appreciated. His comments have unquestionably helped to clarify the text, illustrations and the supplementary data table. Where possible, we have made all his suggested changes, which we detail beneath his specific comments below.

1. Line 38 – move (e.g. NH_4^+) before forms

Done.

2. Line 101 – remove “typically”

Removed.

3. Line 159 – increase to +7 and to +9‰, respectively?

Revised.

4. Line 160 – idem

Revised.

5. Line 190 – Ader et al., 2006 study is on natural samples, I suggest to avoid “experimental” as it can introduce some confusion.

Deleted ‘experimental’, as requested.

6. Line 215 – replace “preserved” by measured

We have replaced ‘reserved’ by ‘original’.

7. Line 371 – we suggest (that?)

Added.

8. Line 376 – at a lower resolution (and magnitude?)

Revised.

9. Final remark: I have some problem with the final part of the final sentence “...when the nitrogen cycle became ecologically modern” - I have the feeling that in the frame of the theory of Evolution (central to ecology) modern can only mean what is observed at the present day or I misunderstand what you mean by modern here. In any case make sure that one could not, somehow, think that the organisms relevant to the N cycle stop evolving sometimes in the Precambrian!

We agree with Dr Thomazo that the characteristics of the modern nitrogen cycle are mainly based on observation. We use 'ecologically' to refer to the ecological importance of each microbial process in the nitrogen cycle, not only the presence of microbial metabolisms. Thus, the phrase 'ecologically modern' refers to the ecological importance of each microbial process. This remains a significant unknown and will require the combination of cutting edge techniques to decipher. We have left this as a closing provocative sentence (only) to entice our "-omics" colleagues and help unveil the evolution of the nitrogen cycle.